# Microbial Biological Control of Fungi Associated with Grapevine Trunk Diseases: A Review of Strain Diversity, Modes of Action, and Advantages and Limits of Current Strategies

**DOI:** 10.3390/jof9060638

**Published:** 2023-05-31

**Authors:** Ouiza Mesguida, Rana Haidar, Amira Yacoub, Assia Dreux-Zigha, Jean-Yves Berthon, Rémy Guyoneaud, Eléonore Attard, Patrice Rey

**Affiliations:** 1E2S UPPA, CNRS, IPREM, Universite de Pau et des Pays de l’Adour, 64000 Pau, France; 2GreenCell: Biopôle Clermont-Limagne, 63360 Saint Beauzire, France

**Keywords:** Botryosphaeria dieback, Esca, Eutypa dieback, biological interactions, plant microbiome, microbial interactions

## Abstract

Grapevine trunk diseases (GTDs) are currently among the most important health challenges for viticulture in the world. Esca, Botryosphaeria dieback, and Eutypa dieback are the most current GTDs caused by fungi in mature vineyards. Their incidence has increased over the last two decades, mainly after the ban of sodium arsenate, carbendazim, and benomyl in the early 2000s. Since then, considerable efforts have been made to find alternative approaches to manage these diseases and limit their propagation. Biocontrol is a sustainable approach to fight against GTD-associated fungi and several microbiological control agents have been tested against at least one of the pathogens involved in these diseases. In this review, we provide an overview of the pathogens responsible, the various potential biocontrol microorganisms selected and used, and their origins, mechanisms of action, and efficiency in various experiments carried out *in vitro*, in greenhouses, and/or in vineyards. Lastly, we discuss the advantages and limitations of these approaches to protect grapevines against GTDs, as well as the future perspectives for their improvement.

## 1. Introduction

As was the case for earlier grapevine health crises at the end of the 19th century with phylloxera, powdery, and downy mildews, the viticulture sector is now confronted with vast upheavals, such as climate change, associated with high societal expectations for an environmentally friendly viticulture, as well as the major crisis of grapevine trunk disease (GTD) epidemics. With regard to GTDs, which re-emerged in the late 1990s, it took a mere two decades for Esca, the most frequent one, to become a subject of major concern for many viticulture regions in Europe and worldwide. GTDs represent a group of vascular diseases caused by fungi affecting grapevine wood, mainly through pruning wounds, and inhabiting the xylem cells in the woody tissue [1,2]. The colonization of this tissue leads to a decline in the plant host because of a loss of the xylem function and subsequent decrease in hydraulic conductivity, causing significant necrosis and decay with time, which ultimately lead to foliar symptoms and grapevine death [3,4,5]. Esca, Botryosphaeria dieback, and Eutypa dieback are the most frequent on mature grapevines; they decrease vineyard longevity, thereby affecting wine quality and causing huge economic losses throughout the viticulture sector [6,7].

Until now, up to 133 fungal species belonging to 34 genera have been associated with GTDs in the literature [8], most of them growing slowly, found alone or together in the same plant, for several years [2,9]. After the infection onset of pathogenic fungi, it takes a long time, usually years, before the appearance of the first foliar symptoms [10]. When the first foliar symptoms occur, they are often linked to the development of rot necrosis in the grapevine trunk or cordons [4]. However, GTDs leave symptoms expressed inconsistently from year to year on individual grapevines [1,2]. The main pathogenic fungi involved in these diseases are *Neofusicoccum parvum*, *Diplodia seriata*, and *Lasiodiplodia theobromae* for Botryosphaeria dieback, *Phaeomoniella chlamydospore*, *Phaeoacremonium minimum*, and *Fomitiporia mediterranea* for Esca, and *Eutypa lata* for Eutypa dieback [1,2].

*Vitis vinifera* cultivars display different levels of tolerance and react with defense mechanisms to cope with the vascular pathogens involved in GTDs [5,10]. The tolerance or susceptibility of grapevine cultivars to vascular fungal pathogens has not yet been fully explained but (i) the small xylem vessel diameter [5,11], (ii) the high levels of phenolic compounds and lignin in the wood, and (iii) the early and rapid induction of defense-related genes with greater accumulation of stilbene compounds and pathogenesis-related proteins [12] have been reported to explain the differences in susceptibility across cultivars.

Previously, sodium arsenate, carbendazim, and benomyl were used to control GTDs. However, the use of these products was banned in early 2000s because of their toxicity toward humans and the environment. To mitigate the economic losses due to GTDs, as no effective control treatments currently exist since, several strategies based on the employment of biological agents, chemical compounds, and cultural practices are used alone or in combination to limit GTD incidence [1,2].

Several methods of control, including cultural practices, chemicals, and biological control products have been tested against grapevine trunk diseases [1]. They can be grouped into preventive and curative methods [6,13]. As for the preventive methods, measures are recommended before, during, and after planting [6,14]. Before planting, it is recommended to use controlled mother vineyards of good quality with limited age, and to avoid the most GTD-susceptible cultivars in the most fertile soils [14]. At planting, it is important to avoid the long immersions of roots in water [14]. After planting and in the vineyards, according to whether the target vines are already affected or not, several prophylactic methods are applied to control GTDs [13,14]. Among them, it is important to take care of the correct training of the trunks by avoiding the short-pruned wounds that can cause drying zones inside the trunk [14]. In addition, the method and time of pruning can affect the susceptibility of wounds to pathogenic fungi [13,14]. Guyot-Poussard is the most used pruning system as it ensures an optimal flow of sap [13]. However, the use of such a system is still not fully understood or justified, due to a lack of evidence of efficiency in relevant experimental trials [13]. Lecomte et al. (2012) recommended a pruning period in late winter, particularly to prevent *Eutypa* [15]. The protection of pruning wounds using natural or chemical products is another method used to limit wound infection by pathogens [8,16].

As for the curative methods in GTD-affected vineyards, “remedial surgery” is applied to eliminate by pruning the symptomatic woody parts from affected vines (cordons and/or trunks), until healthy wood is left [8,13]. If the majority of the vine trunk exhibits internal symptoms of GTD, the technique used is the trunk renewal [8]. In some countries, trunk surgery or “curettage” is another practice used to remove the rotten tissues in the trunk of GTD-affected vines using electric handsaws [13].

All these management strategies may help to prevent GTDs. However, GTD control is still challenging and problematic because of the rarity of efficient strategies, as well as the complexity of the diseases with a high diversity of biotic and abiotic factors involved in the different disease stages. Wound protection remains the most effective technique for limiting the dissipation of pathogens. Thus, the search for effective strategies for the protection of wounds, notably via biological control, is essential for the management of GTDs.

Biological control is a promising sustainable alternative approach to fight GTD-causing fungal pathogens; during the last decade, about 1600 microorganisms with potential biocontrol activities (MBCAs) were investigated. Most of the studies were carried out with bacterial and fungal strains, but a few oomycetes and actinobacteria were also used to biocontrol GTD pathogens [1]. These MBCAs stopped and/or destroyed the pathogenic fungi through a number of direct or indirect mechanisms of action [17,18]. Direct interactions occur when there is competition for spaces and nutrients, the production of siderophores and hydrolytic enzymes, parasitism, or antibiosis [18]. On the other hand, indirect mechanisms mainly consist of the induction of defense mechanisms in the plant further to its colonization by a biocontrol agent (Figure 1).

In this review, we provide an overview of the current knowledge about the microbiological control agents used to manage the main pathogens involved in GTDs. We highlight the empirical evidence on their potential efficiency and mechanism of action, and we outline the current practices used to manage GTDs.

## 2. Biocontrol of Botryosphaeria Dieback

Botryosphaeria dieback is associated with several Botryosphaeriaceae species [2]. Around 26 different Botryosphaeriaceae taxa have been found in vineyards of several countries, but *Neofusicoccum parvum*, *Diplodia seriata*, *Phaeoacremonium minimum*, *Lasiodiplodia theobromae*, *Neofusicoccum australe*, *Neofusicoccum luteum*, and *Botryosphaeria dothidea* are the most widespread and aggressive species associated with Botryosphaeria dieback [8,19,20,21]. They cause shoot dieback, cankers, central necroses in wood, and/or grapevine dieback [22]. The species within the genera *Lasiodiplodia* and *Neofusicoccum* are the fastest wood-colonizing fungi [8,23].

Most of this section is dedicated to biocontrol of the three most studied Botryosphaeriaceae species, *N. parvum*, *D. seratia*, and *L. theobromae*, before ending with three papers aimed at controlling *N. australe* or other Botryosphaeria dieback-associated fungi.

### 2.1. Biological Control of Neofusicoccum parvum

Regarding the pathogenicity of *N. parvum*, Pitt et al. (2013) reported that *N. parvum* is one of the most virulent species associated with Botryosphaeria dieback, according to the lesion length they produce on mature wood tissue [24]. During grapevine colonization, *N. parvum* produces phytotoxic metabolites with low molecular weight, including (3R,4R)-(−)-4-hydroxy-mellein and its stereoisomer (3R,4S)-(−)-4-hydroxy-mellein, (−)-(R)-mellein, (−)-terremutin, isosclerone, and tyrosol [25,26,27,28]. *N. parvum* also produces hydrophilic high-molecular-weight exopolysaccharides with phytotoxic activities [25,26]. The phytotoxic activities of these secondary metabolites have been elucidated, but their contribution to the development of Botryosphaeria dieback symptoms is still unknown [27,28]. In addition, it was reported that *N. parvum* produces extracellular proteins with enzymatic activities involved in wood degradation such as hydrolases and oxidoreductases, which are likely involved in cell-wall and lignin degradation [19]; they are considered virulent factors responsible for pathogenicity [29].

Therefore, *N. parvum* is a key target to develop biocontrol products against Botryosphaeria dieback. From this perspective, multiple fungus isolates have been evaluated for their potential antagonistic activity against *N. parvum* such as *Chaetomium* sp., *Cladosporium* sp., *Clonostachys rosea*, *Epicoccum* spp., *Epicoccum nigrum*, *Fusarium proliferatum*, *Purpureocillium lilacinum*, and *Trichoderma* spp. [1,30,31,32].

#### 2.1.1. Biocontrol Using *Trichoderma*

##### *Trichoderma* spp.—*N. parvum* Interactions *In Vitro*

*Trichoderma* spp. showed high efficiency in wound protection against all GTD pathogens [28]. In 2020, Úrbez-Torres et al. tested in dual culture the antagonistic capabilities of 16 *Trichoderma* strains isolated from southern Italy against *N. parvum*. The highest percentage inhibition of radial mycelial growth of *N. parvum* (74.3%) was obtained with the strain *T. koningiopsis* PARC1024 that was isolated from *Prunus persica* [33]. Ten Trichoderma strains were assessed for their antagonistic activity against *N. parvum in vitro* by Kotze et al. (2011); two *T. atroviride* strains, coded USPP-T1 and USPP-T2, were isolated from *V. vinifera* in South Africa, and there were eight commercial strains, i.e., three *T. atroviride*, coded AG3, AG5, and AG8, and five *T. harzianum*, coded AG2, AG11, Agss28, Biotricho, and Eco77. All these strains were able to overgrow the pathogens *in vitro*, and microscopic observations revealed coiling or hyphal adhesion between the pathogen’s hyphae and the Trichoderma’s hyphae (for strains AG3, AG5, AG11, USPP-T1, and USPP-T2) [34]. Kotze et al. (2011), suggested mycoparasitism and competition for nutrients as mechanisms of action of these strains.

As for *Trichoderma* endophytes, the growth of *N. parvum* was significantly reduced by about 80% using the endophytic strains *T. atroviride* (ATCC 74058) and *T. harzianum* (ATCC 26799) on PDA. These two strains did not show any ability to outcompete this pathogen on carbon or nitrogen sources, although *T. harzianum* had some niche overlap with it [35]. Blundell et al. (2021) reported for the first time the efficiency of a grapevine sap *T. hamatum* strain against *N. parvum in vitro*.

Recently, Kovács et al. (2021) examined the potential ability of two *Trichoderma* strains isolated from cordon wood of the grapevine cultivar Furmint in Hungary to inhibit the growth of *N. parvum in vitro*. The two isolates, identified as *T. afroharzianum* (TR04) and *T. simmonsii* (TR05), showed high potential against this pathogen with biocontrol indices of 95.19% and 90%, respectively. In dual culture, *T. afroharzianum* (TR04) and *T. simmonsii* (TR05) overgrew the *N. parvum* colony, and their hyphae coiled and penetrated the ones of the pathogens [31], which is a sign of mycoparasitism [36]. According to Pollard-Flamand et al. (2022), 26 *Trichoderma* strains obtained from grapevine roots and the basal end of either rootstock or self-rooted vines in British Columbia significantly inhibited the growth of *N. parvum in vitro*. They belonged to seven species: *T. asperelloides*, *T. atroviride*, *T. harzianum*, *T. koningii*, *T. tomentosum*, *T. canadense*, and *T. viticola*.

##### *Trichoderma* spp.—*N. parvum* Interactions *In Planta*

Under controlled conditions, on detached grapevine cane, one strain of *T. atroviride*, one of *T. paratroviride*, and one of *T. guizhouense* effectively protected pruning wounds against *N. parvum* for at least 21 days after treatment. *T. paratroviride* PARC1012 gave a mean percentage of disease control greater than 90% only 1 day after treatment [33]. These authors hypothesized that the mode of action of *Trichoderma* spp. to protect pruning wounds against *N. parvum* was competition for nutrients and space [33]. *T. canadense*, *T. viticola*, *T. harzianum*, *T. atroviride*, *T. asperelloides*, and *T. koningii* were tested for their ability to protect pruning wounds against *N. parvum* on detached cane assays under controlled greenhouse conditions; all these strains were effective in protecting from the pathogenic infection. Higher protection was obtained with the strains of *T. asperelloides*, *T. atroviride*, and *T. canadense*, which provided 96–100% pruning wound protection for up to 21 days after treatment [37].

In the vineyard, Kotze et al. (2011) evaluated the ability of 10 Trichoderma strains sprayed on fresh pruning wounds in a South African vineyard. The results showed that the strain *T. atroviride* USPP-T1 was the most efficient and reduced the incidence of *N. parvum* by 80% when challenged 7 days after treatment by the pathogen [34].

#### 2.1.2. Biocontrol Using Other Fungal Genera

*In vitro*, it was reported that *Chaetomium* sp. showed a significant reduction in *N. parvum* growth *in vitro* [1,30] with a reduction by 86.75% after 21 days of dual culture on an agar medium based on 200 g/L grapevine dormant cutting. These authors hypothesized that the antagonistic activity of *Chaetomium* sp. against *N. parvum* was due to mycoparasitism because it grew slowly and inhibited the pathogen until colony contact [30].

*Cladosporium* sp. obtained from sprouts of asymptomatic grapevine showed interesting antagonist activity against *N. parvum* in the agar medium based on 200 g/L grapevine dormant cutting. The growth of this pathogenic fungi was reduced by 34.26% in the presence of *Cladosporium* sp. Its antagonistic activity was presumably due to two modes of action: (i) antibiosis, since, in the coculture test, *N. parvum* growth terminated before direct contact of the two colonies; (ii) the highest sporulation rate of *Cladosporium* sp., because, in the dual-culture assay, the growth inhibition of *N. parvum* was related to the strong sporulation of *Cladosporium* sp. and not to the rapid growth of its mycelium [30].

*Epicoccum* is a genus of Ascomycetes associated with the wood mycobiome of grapevines with known biocontrol potential [38,39]. The strain *Epicoccum nigrum* R29.1, isolated from the root of asymptomatic grapevine, was tested *in vitro* for its potential to inhibit the growth of *N. parvum*, but no significant results were obtained [30].

The antagonistic activity of *C. rosea* against GTD-associated fungi has been investigated recently [30,40]. Silva-Valderrama et al. (2021) studied three *C. rosea* strains *in vitro* and *in planta* and suggested that *C. rosea* strains were promising biocontrol agents of GTD pathogens. These *C. rosea* strains were isolated from asymptomatic Cabernet Sauvignon and Chardonnay commercial vineyards in Chile. Two strains (*C. rosea* CoR2.15 and *C. rosea* R36.1) were root endophytes, and one (*C. rosea* CoS3/4.24) was from the rhizosphere. When these three strains were tested to control *N. parvum in vitro*, all of them inhibited over 98% of pathogen growth on day 21 [30]. For two strains, there was direct contact between colonies, and light microscope observation revealed hyphal coiling in the confronting zone of the two mycelia, suggesting mycoparasitism as the mechanism of action. As for the third strain, the *N. parvum* growth inhibition was without physical contact between the colonies, and its growth was terminated in correspondence with the halo surrounding it [30]. There were changes in the colony morphology of *N. parvum*, which turned into several flat independent colonies with undulate margins in contact with secondary metabolites secreted by *C. rosea* CoS3/4.24. In this case, it was suggested that the antagonistic activity of *C. rosea* was due to the secreted antibiotic compound [30].

*Purpureocillium lilacinum* has antibacterial, antimalarial, antifungal, antiviral, and antitumor activities, and it is also known for its toxic activities against phytopathogens, notably *Phytophthora capsica* [41]. Recently, it was shown that *P. lilacinum* inhibited *N. parvum in vitro* without evident physical contact between colonies, suggesting the secretion of secondary metabolites [30].

*Aureobasidium* spp. isolated from grapevine canes and sap, as well as one strain identified as *A. pullulans*, were inefficient against *N. parvum* when tested *in vitro* [42]. It was shown that *Lecanicillium lecanii* (ATCC 46578) caused a reduction by 15% in *N. parvum* growth *in vitro*, acting via direct antagonism. According to Wallis et al. (2021), it also effectively some niche overlap with this pathogen.

*In planta*, Silva-Valderrama et al. (2021) performed an assay on annual detached shoots to study the antagonistic activity of the three *C. rosea* strains mentioned above against *N. parvum*. These strains had a good efficiency to inhibit *N. parvum*, with the endophytic isolates (*C. rosea* CoR2.15 and *C. rosea* R36.1) showing a better inhibition of *N. parvum* in grapevine woody shoots compared with the rhizospheric strain, *C. rosea* CoS3/4.24 [30]. Mondello et al. (2019) reported that, under greenhouse-controlled conditions, and in a vineyard planted in 1997 in France with the Mourvèdre/3309 cultivar, *Fusarium proliferatum* limited the development of *N. parvum* by priming plant defense response when this pathogen was inoculated 7 days after treatment with the *Fusarium* [32].

#### 2.1.3. Biocontrol Using Oomycetes

*Pythium oligandrum* is a rhizospheric, nonpathogenic oomycete that colonizes the root system of many cultivated plant species, including grapevine [43,44]. The biological control exerted by *P. oligandrum* is due to direct effects on the pathogens (i.e., via mycoparasitism or antibiosis) and/or indirect effects by resistance induction and growth promotion of the plant [45]. Daraignes et al. (2018) carried out a 2 year study demonstrating that *P. oligandrum* root colonization reduced the wood necrosis length caused by *N. parvum* in young, grafted cuttings of Cabernet Sauvignon grapevine. Because the pathogen and *P. oligandrum* colonized different plant organs and never came into contact, these authors assumed that induction of the grapevine defense system was the mode of action of *P. oligandrum*. Under similar greenhouse conditions, Yacoub et al. (2020) showed that root system colonization by *P. oligandrum* was associated with the reduction in wood necrosis caused by *N. parvum* in rooted cuttings of Cabernet Sauvignon grapevine [43]. These authors also studied the expression of 62 genes involved in grapevine defense pathways and observed that the priming of certain genes occurred at early stages, 14 days after the pathogen inoculation. They highlighted the upregulation of PR protein genes, e.g., *PR1*, a marker of the salicylic acid pathway and antifungal activity, *GLU* and *PR2* encoding 6-1,3-glucanase, *PR4bis* encoding chitinase, and *PR14* involved in the defense signaling pathway, as well as those involved in cell-wall reinforcement (e.g., *CAD* and *CAGT*), the indole signaling pathway (e.g., *HSR203J*, *CHORM*, and *CHORS2*), and hormone signaling pathways (e.g., *EDS1*, *ACO1*, *SAMT1*, and *WRKY2*), and genes affecting the salicylic acid pathway (e.g., *SAMT1* and *EDS1*). No synergetic effect between *P. oligandrum* and a bacterial strain with potential biocontrol activity, *Pantoea agglomerans* or *Bacillus pumilus*, was observed by Daraignes et al. (2018) in their greenhouse experiment.

#### 2.1.4. Biocontrol Using Bacteria

The bacterial biocontrol of Botryosphaeria dieback pathogens has been explored, mainly targeting the pathogen *N. parvum* [1]. Strains belonging to *Bacillus* spp. isolated from healthy vineyards were highly efficient in protecting pruning wounds against various GTD pathogens *in vitro*, in the nursery/greenhouse, and even in the field [28,42]. *Bacillus* spp. antagonized GTD pathogens via various modes of actions such as antibiosis, competition for nutrients, activation of plant defense system, and detoxification of pathogen toxins [28,42]. Strains of *Bacillus subtilis* were described as promising plant protectors against many fungal pathogens, including in grapevine against pathogens causing wood staining [46,47,48].

In an *in vitro* study with *B. subtilis*, Kotze et al. (2011) used a strain isolated from the woody tissue of grapevine wood arms (Chenin Blanc cultivar) that expressed Eutypa dieback symptoms in South Africa. In the inhibition zone between *N. parvum* colonies and the *B. subtilis* strain, light swelling and malformation of the pathogen hyphae was observed, likely due to antibiotic molecule production by the bacteria [34].

In greenhouse studies with *B. subtilis* on young grapevine plants, a *B. subtilis* strain (coded PTA-271) isolated from the rhizosphere of healthy Chardonnay grapevines in Champagne (France) was able to use both indirect and direct mechanisms to protect grapevine cuttings against *N. parvum*. Furthermore, the inoculation of grapevine cuttings with this PTA-271 bacterial strain in the soil for 1 month, then with the pathogen, significantly enhanced systemic grapevine immunity by priming the expression of *PR2*, encoding enzymes involved in abscisic acid biosynthesis [28]. This *B. subtilis* strain also triggered the expression of salicylic acid- and jasmonic acid-responsive genes involved in the detoxification process of key aggressive phytotoxins produced by *N. parvum*, i.e., (−)-terremutin and (R)-mellein. Because this detoxifying process is more active in a nutrient-rich medium for (−)-terremutin, but not for (R)-mellein, Trotel-Aziz et al. (2019) suggested that (R)-mellein was probably metabolized directly, while (−)-terremutin required a co-substrate to be co-metabolized. This bacterium also acted directly on the pathogen, as shown by its fungistatic effect on *N. parvum* hyphae [28]. 

With respect to assays in the field, in South African vineyards, after 8 months of treatment with *B. subtilis*, a reduction by 16.5% in the incidence of this pathogen was observed by Kotze et al. (2011); however, this decrease was not significantly different from observations on unprotected wounds.

Regarding other *in vitro* assays with *Bacillus* and other bacteria species, Blundell et al. (2021) investigated the *in vitro* ability of *Bacillus velezensis* to inhibit *N. parvum*. Given that they observed a small zone of inhibition in the dual-culture assay and in the volatile assay corresponding to 10% growth inhibition, they concluded that a volatile antibiotic was produced [42]. The endophytic *Bacillus* sp. 3R1, *Brevibacillus* sp. 3Y41, and *Bacillus* sp. 3R4 strains, isolated from a 3 year old grapevine cultivar Corvina in Italy, were able to inhibit the growth of *N. parvum in vitro*. The *B. methylotrophicus* 3R1 strain expressed the strongest antifungal activity [49]. Another bacterial species, *Pseudomonas protegens* MP12, isolated from Italian forest soil, and the strain *P. protegens* DSM 19095T significantly inhibited the mycelial growth of *N. parvum* when assessed *in vitro* [50]. Two strains of *P. agglomerans* (S1 and S3) and one of *Paenibacillus* sp. (S19), isolated from grape berries and wood tissue, respectively, inhibited the growth of *N. parvum* via the production of phenylethyl alcohol, an antifungal volatile compound, while *Paenibacillus* sp. directly inhibited *N. parvum* via antibiosis [22].

In a recent study, Bustamante et al. (2022) evaluated the antagonistic activity of 1344 endophytic and rhizospheric bacterial isolates against *N. parvum*. These bacterial strains were isolated from different grapevine cultivars in California. The result showed that 172 isolates inhibited *N. parvum* growth by more than 40% in the dual-culture assay. These bacteria belonged to the species *B. velezensis* (154 isolates), *Pseudomonas* spp. (12 isolates), *Serratia plymuthica* (two isolates), and other genera (four isolates) [51]. In the same study, it was reported that *B. velezensis* (two strains), *Pseudomonas chlororaphis* (two strains), and *Serratia plymuthica* (two strains) reduced the mycelial growth of *N. parvum* via their agar-diffusible metabolites. However, they gave a low inhibition on *N. parvum* mycelial growth via the production of volatile organic compounds [51].

With regard to *in planta* assays, several bacteria were tested to fight *N. parvum* infection on young plants, but no protection was observed with *Acinetobacter radioresistens*, *B. firmus*, *B. ginsengihumi*, *B. licheniformis*, *B. pumilus*, *Brevibacillus reuszeri*, *Curtobacterium* sp., *Enterobacter cowanii*, *Paenibacillus barengoltzii*, *Paenibacillus illinoisensis*, *Paenibacillus polymyxa*, *Paenibacillus turicensis*, and *Xanthomonas* sp. [1,52]. However, other bioassays were more positive in terms of biocontrol protection, such as the one conducted under greenhouse conditions by Haidar et al. (2021). They showed that *Enterobacter* sp. S24 and *B. firmus* S41, isolated from grapevine wood, reduced the internal necrosis lesion length caused by *N. parvum*. These authors demonstrated that, in addition to the *in vitro* inhibition of *N. parvum* by *P. agglomerans* S1 and *Paenibacillus* sp. S19, these bacterial strains were able to protect young grapevine from *N. parvum* infection via an indirect mechanism, i.e., induction of plant resistance. They showed also that, among the various modes of application of these potential biocontrol agents on plants, preventive inoculation on the stem was the most efficient in controlling *N. parvum* [22]. Other studies showed that *P. agglomerans* reduced the length of necrosis caused by *N. parvum* by 30% on grafted and by 32.3–43.5% on nongrafted cutting stems of Cabernet Sauvignon cultivar under greenhouse conditions [52,53].

#### 2.1.5. Biocontrol Using Actinobacteria

Regarding actinobacteria, 40 endophytic actinobacteria isolated from grapevine cv. Sauvignon Blanc and identified as *Streptomyces* spp. were tested for their antagonistic activity against *N. parvum*. Among them, 29 strains highly or moderately inhibited the growth of this pathogen *in vitro* [54]. 

A wide variety of MBCAs have been tested *in vitro* and *in planta* against *N. parvum* with *Trichoderma* spp. as the most tested biocontrol agent. The majority of *Trichoderma* strains showed a very good efficiency *in vitro*; some of these strains showed a constant efficiency in the greenhouse and the vineyard. Generally, a constant efficiency was observed for other fungal isolates, such as *C. rosea* that gave a good antagonistic activity *in vitro* and *in planta* under controlled conditions, and for *F. proliferatum* that inhibited the pathogen both *in planta* and in the vineyard. The oomycete *P. oligandrum* also showed a good efficiency under greenhouse conditions, but studies in vineyards are needed to confirm the efficiency achieved *in vitro* and in greenhouses. Regarding bacterial strains, most of them were highly effective *in planta* and/or *in vitro*. Strikingly, only one bacterial strain was tested in the vineyard, but its effectiveness was inconsistent.

### 2.2. Biological Control of Diplodia seriata

Although less virulent than *N. parvum*, *Diplodia seriata* is one of the most aggressive species isolated from diseased grapevines worldwide [19,21,55]. Various MBCAs have been tested against this pathogen *in vitro*, in the nursery, and in the field to protect wounds. As usual, depending on the strains, various levels of protection were obtained.

#### 2.2.1. Biocontrol Using *Trichoderma*

*In vitro Trichoderma* spp. strains isolated from Southern Italy by Úrbez-Torres et al. (2020) overgrew *D. seriata* mycelium. *T. atroviride* was the most efficient with a percentage radial growth inhibition of 69.6%. According to Kovács et al. (2014), 10 *Trichoderma* spp. isolated from the grapevine trunk also overgrew *D. seriata* mycelium *in vitro* [56]. *T. atroviride* strains, USPP-T1 and USPP-T2, had an inhibitory effect on *D. seriata*, via the production of secondary metabolites [34]. Indeed, an inhibitory zone between the colonies of these two strains and that of *D. seriata* was observed, with hyphal disintegration of the pathogen. Kovács et al. (2021) tested two *Trichoderma* strains identified as *T. afroharzianum* (strain TR04) and *T. simmonsii* (strain TR05) isolated from grapevine cordon wood. These two strains overgrew the pathogen colony, along with hyphal coiling and penetration in pathogen hyphae, suggesting mycoparasitism as the mechanism of action [31]. Recently it was reported that 26 isolates of *Trichoderma* including species *T. asperelloides*, *T. atroviride*, *T. harzianum*, *T. koningii*, *T. tomentosum*, *T. canadense*, and *T. viticola* significantly inhibited the growth of *D. seriata in vitro* [37].

Regarding *Trichoderma* endophytes, Silva-Valderrama et al. (2021) reported that the *Trichoderma* sp. strain Altair isolated from grapevine inhibited *D. seriata* growth as early as 7 days *in vitro*. Light microscope observations revealed that this *Trichoderma* strain produced hyphal coils when it interacted with two *D. seriata* colonies, suggesting mycoparasitism as a mode of action [30]. According to Wallis (2021), two other endophytes, *T. atroviride* (ATCC 74058) and *T. harzianum* (ATCC 26799), were efficient in reducing *D. seriata* growth by over 75%. The *T. harzianum* strain was qualitatively the most efficient in controlling the pathogen as it outcompeted *D. seriata* for carbon and nitrogen sources [35]. In the literature, strains belonging to the *Trichoderma* genus, i.e., *Trichoderma* sp., *T. longibrachiatum*, *T. harzianum*, *T. atroviride*, *T. afroharzianum*, and *T. simmonsii*, were highly efficient in competing *in vitro* against *D. seriata*, but this was also observed with strains from other genera such as *C. rosea, F. proliferatum*, and *Cladosporium* sp. [1,30,31].

*In planta* experiments were usually conducted with *Trichoderma* spp. to protect pruning wounds. For instance, *T. paratroviride*, *Trichoderma* sp., and two strains isolated from *P. persica*, i.e., *T. koningiopsis* and *T. guizhouense*, controlled *D. seriata* infection by 89–94% on pruning wounds when challenged with the pathogen at least 21 days after treatment [33]. Seven *Trichoderma* spp. isolates were tested on plated detached grapevine canes under controlled greenhouse conditions to protect pruning wounds from *D. seriata*; all strains showed moderate or high ability to protect pruning wounds from this pathogen. *T. harzianum*, *T. atroviride*, and *T. asperelloides* were the most effective with a mean percentage disease control of 97–100%, 21 days after treatment [37]. Kotze et al. (2011), in an experiment in South African vineyards, interestingly observed that the *T. atroviride* strain USPP-T1 reduced the incidence of *D. seriata* by 85% after 8 months. In parallel, these authors evaluated eight strains of three commercial products belonging to the species *T. atroviride* (AG3, AG5, and AG8) and *T. harzianum* (AG2, AG11, Agss28, Biotricho, and Eco77). These strains, except one, were able to overgrow *D. seriata* and operated through mycoparasitism. The three bioproducts were able to reduce *D. seriata* incidence on pruning wounds under field conditions [34].

#### 2.2.2. Biocontrol Using Other Fungal Genera

Fungi from the *Chaetomium*, *Cladosporium*, *Clonostachys*, *Fusarium*, and *Lecanicillium* genera have been tested against *D. seriata*, some of which are endophytes. Silva-Valderrama et al. (2021) reported that three *C. rosea* strains completely overgrew *D. seriata in vitro* by day 21, but they had various modes of action. The antagonistic activity of the strain *C. rosea* CoS3/4.24, isolated from the grapevine rhizosphere, was associated with both a secreted antibiotic compound and mycoparasitism. Indeed, the secreted metabolites of *C. rosea* CoS3/4.24 reduced the growth of *D. seriata* by 47.2% and changed its colony morphology, whereas hyphal coiling, associated with mycoparasitism, was observed in the confronting zone of the two fungal cocultures. For the two other strains, the antagonistic activity of *C. rosea* R36.1 was due only to mycoparasitism, while that of *C. rosea* CoR2.15 was due only to antibiosis. In *in planta* trials, the strain CoS3/4.24, with two modes of action, was used, and a significant growth inhibition of *D. seriata* was observed in all assays [30].

Other endophyte strains from various fungal genera have displayed antagonistic activity *in vitro* against *D. seriata*. In dual culture, *Cladosporium* sp. acted on *D. seriata* via antibiosis, whereby the growth inhibition (42.46%) of *D. seriata* operated through metabolites secreted by the antagonist [30]. For *Chaetomium* sp., its mechanism of action was related to a slow mycoparasitism, as its hyphae penetrated and coiled around those of *D. seriata* on day 30 of coculture [30]. Wallis (2021) reported that the endophytic strain of *Lecanicillium lecanii* (ATCC 46578) reduced *D. seriata* growth *in vitro* by about 20%, via direct antagonism and competition for carbon and nitrogen sources [35]. In the experiment of Blundell et al. (2021), two other *A. pullulans* strains (coded UCD 8189 and 8344), isolated from grapevine sap and cane tissue from healthy Chenin Blanc cultivar, caused significant inhibition of *D. seriata* radial mycelial growth, but no inhibitory effect was obtained in the volatile assay.

In the study of Pinto et al. (2018), conducted *in planta* on grapevine cuttings cv. Chardonnay, another fungal endophyte, A. *pullulans* strain Fito_F278, isolated from leaves of *V. vinifera* in Portugal, was reported to have an indirect effect on *D. seriata* growth. This strain promoted the induction of some plant defense responses in cutting plants, 1 week after *D. seriata* inoculation. For instance, the expression of genes encoding plant defense proteins, such as PR protein 6 (*PR6*) and β-1,3-glucanase (*Gluc*), were upregulated [57]. In addition to plant defense induction, these authors suggested that this *A. pullulans* Fito_F278 strain was also able to compete with GTD fungi in the field, as it colonized the grapevine at an endophyte and epiphyte level. *F. proliferatum* was reported to be a pathogen for several crops, but it had an antagonistic effect on the oomycete *Plasmopara viticola*, the causative agent of grapevine downy mildew [58]. It limited the growth of *D. seriata in vitro* through antibiosis and direct antagonism [32].

#### 2.2.3. Biocontrol Using Bacteria

Regarding bacteria, experiments with strains from *Bacillus* and genera isolated from grapevine organs have been conducted. A *B. subtilis* strain isolated from the arm’s wood of the cultivar Chenin Blanc that expressed Eutypa symptoms was highly efficient *in vitro* against *D. seriata* [34]. Its antagonistic activity was attributed to antibiotic compound production and diffusion, causing hyphal malformation such as swelling [34]. In the field, *B. subtilis* reduced the incidence of *D. seriata* in fresh pruning wounds of Chenin Blanc and Merlot grapevine cultivars when the pathogen was inoculated 7 days after the biocontrol treatment [34]. Blundell et al. (2021) isolated two *B. velezensis* strains from sap and cane tissue of grapevine, which significantly inhibited *D. seriata* in dual culture. Bustamante et al. (2022) showed that 172 endophytic and rhizospheric bacterial isolates, including *B. velezensis* (154 isolates), *Pseudomonas* spp. (12 isolates), *Serratia plymuthica* (two isolates), and four isolates from other genera, inhibited the growth of *D. seriata* by more than 40% *in vitro*. However, when a bacterial strain of *Burkholderia phytofirmans* was used in an *in planta* bioassay, it had no efficiency in inhibiting *D. seriata* infection [1]. To the best of our knowledge, no actinobacteria or oomycete strains have been tested against *D. seriata*.

Overall, among the fungal strains, there was a high and consistent efficacy of *Trichoderma* isolates tested *in vitro* and in greenhouse/field conditions. As for bacteria, most of them were highly efficient *in vitro* and in greenhouse conditions. However, only one bacterial isolate was tested *in planta*, and its efficiency was not maintained over time. Therefore, more studies are required to understand and evaluate the ability of bacteria in more realistic conditions. To the best of our knowledge, no studies are available on the antagonistic activity of oomycetes or actinobacteria against *D. seriata*.

### 2.3. Biological Control of Lasiodiplodia theobromae

*Lasiodiplodia theobromae* which is frequently found in tropical and subtropical regions, is the most representative and aggressive species of the genus *Lasiodiplodia* involved in grapevine Botryosphaeria dieback [59,60]. The taxonomy of *Lasiodiplodia* was recently revised. As a consequence, fungal isolates previously reported as *L. theobromae* were reclassified as new species. A number of species were then reduced to synonyms [61]. Potential bacterial and fungal MBCAs were tested *in vitro* and *in planta* against this pathogen.

#### 2.3.1. Biocontrol Using Fungi

Strains from various species of the genus *Trichoderma* have been assessed to control *L. theobromae*, such as *T. atroviride*, *T. harzianum*, *T. koningiopsis*, *T. asperellum*, and *T. asperelloides* [62]. *In vitro* experiments carried out by Kotze et al. (2011) showed that *Trichoderma* spp. strains had various modes of action. Indeed, one *T. harzianum* strain (i.e., AG2) acted on *L. theobromae* via mycoparasitism, while another *T. harzianum* strain (i.e., Biotricho) and two of *T. atroviride* (i.e., AG3 and AG5) likely had both antibiosis and mycoparasitism as modes of action. However, in dual culture, the targeted fungus was able to defend itself, as likely seen when *T. atroviride* strain AG8 and those of *L. theobromae* inhibited each other [34]. In another experiment, strains of *T. harzianum*, *T. asperelloides*, *T. asperellum*, and *T. koningiopsis* were substantial antagonists to *L. theobromae* 14 days after dual inoculation [62]. In the same study, the strain *T. asperelloides*, coded 02/03, showed endophytic penetration capacity in grapevine cane; in an *in planta* assay on healthy Niagara Rosada grapevine shoots, this strain had a preventive and curative capability to control *L. theobromae*, by protecting the pruning wounds from *L. theobromae* at 20 days post inoculation [62].

In the field, the species *T. atroviride* was identified as a promising candidate to protect pruning wounds against *L. theobromae* [1,34]. Kotze et al. (2011) showed that two *T. atroviride* strains obtained from a vineyard in South Africa were effective enough to reduce the incidence of *L. theobromae* by 92% when the pathogen was applied 7 days after biocontrol treatment. Light microscope observation revealed a coiling between *T. atroviride* and *L. theobromae*’s hyphae, suggesting mycoparasitism as the mechanism of action. Only another endophyte fungal species, i.e., *Epicoccum purpurascens*, displayed efficiency in controlling *L. theobromae in vitro* [1,63].

#### 2.3.2. Biocontrol Using Bacteria

*B. subtilis* and *Bacillus* sp. (AG1) were the only bacterial antagonists to be tested against *L. theobromae* [1,34,64]. In a dual-culture assay, *B. subtilis* inhibited *L. theobromae* growth, and an inhibition zone was observed, associated with swilling and malformation of the pathogen’s hyphae. Kotze et al. (2011) suggested that this effect could be attributed to antibiotic substance production. For a *B. subtilis* (AG1) isolated from grape wood tissues affected by Esca (reclassified as *B. amyloliquefaciens* in 2012 [65]), Alfonzo et al. (2009) showed that the metabolites produced were, in part, responsible for its inhibitory effect against *L. theobromae*. Under field conditions, pruning wounds treated with *B. subtilis* showed significantly lower *L. theobromae* incidence 8 months after its inoculation [34].

Overall, only *Trichoderma* strains and one isolate of *Epicoccum purpuascens* have been studied *in vitro* to control *L. theobromae*. Among the *Trichoderma* isolates, some efficiently controlled the pathogen *in vitro* and under field conditions. Only two bacterial strains have been tested, and both of them were effective. One of them (*B. subtilis*) showed great efficiency in vineyards to protect pruning wounds.

### 2.4. Biocontrol of Neofusicoccum australe and Other Botryosphaeria Dieback-Associated Fungi

Experiments with fungi and bacteria with potential biocontrol properties were carried out to fight *Neofusicoccum australe*, one of the most virulent species associated with botryosphaeria dieback [66].

To control *N. australe*, Kotze et al. (2011) used *Trichoderma* strains, from commercial products, i.e., *T. harzianum* and *T. atroviride*, or isolated from grapevine in South Africa, i.e., two *T. atroviride* (USPP-T1 and USPP-T2) strains, and one of *B. subtilis*. These strains inhibited the pathogen *in vitro*, by stopping the pathogen growth when the two colonies entered in contact (*T. harzianum* Eco 77) and/or by establishing an inhibiting zone between the colonies (*T. harzianum* ag 11). The pathogen growth was first stopped then overgrown (USPP-T1 and USPP-T2) or immediately overgrown, as observed with the strains *T. harzianum* AG2, AG11, AG28, and Biotricho, as well as *T. atroviride* AG3, AG5, and AG8 [34]. Under field conditions, in order to protect pruning wounds from *N. australe*, the *T. atroviride* strain coded USPP-T1 was the most efficient with 78% reduction in incidence 8 months after pathogen inoculation [34].

As for Botryosphaeria dothidea, *Botryosphaeria stevenssi* (currently named *Diplodia mutila*), *Diplodia corticola*, *Neofusicoccum luteum*, *Neofusicoccum mediterraneaum*, and other pathogens associated with Botryosphaeria dieback, *T. atroviride* controlled almost all pathogens *in vitro*, while *T. gamsii* was effective against *B. stevenssi* (*D. mutila*) in *in vitro* studies [1]. *T. atroviride* controlled *N. luteum* and *N. mediterraneaum* efficiency in *in planta* experiments [1].

## 3. Biocontrol of Esca

Esca is a white rot disease caused by a set of fungal Ascomycetes and Basidiomycetes on the wood of grapevines [1,2]. The colonization of grapevine trunk and cordon woody tissues by fungi, mainly *Phaeomoniella chlamydospora*, *Phaeoacremonium minimum*, and *Fomitiporia mediterranea*, causes various types of necrosis. Bruno et al. (2020) suggested that these three fungi disturb various morphological, physiological, and biochemical functions in grapevine during the vegetative period, subsequently affecting bleeding xylem sap and leaves, flux, dynamic viscosity, and growth regulator activity. They also alter grapevine phenol metabolism according to Bruez et al. (2021). In the literature, it is assumed that Esca results from the successive and coordinated action of these pathogenic fungi; *P. chlamydospora* reduces plant resistance due to its toxic activity, *P. minimum* affects cell wall integrity through its enzymatic activity, and, at the last stage of the disease, *F. mediterranea* takes advantage of the cellular degradations caused by the previous fungi to cause complete degradation of wood tissues, resulting in white rot necrosis formation [1,67,68,69,70]. Recently, Haidar et al. (2021) showed for the first time that the fungal ability to degrade wood was strongly influenced by wood-inhabiting bacteria. They demonstrated that a cellulolytic and xylanolytic *Paenibacillus* sp. strain displayed a synergistic interaction with *F. mediterranea* to enhance wood degradation structures [71].

### 3.1. Biological Control of Phaeomoniella chlamydospora

Many MBCAs have been used to control *P. chlamydospora*, and antagonistic species belonging to the *Trichoderma* genus have been tested as the most effective against this pathogen [1]. For instance, *T. atroviride, T. harzianum, T. hamatum, T. longibrachiatum, T. gamsii*, and *Trichoderma* sp., when tested *in vitro* against this pathogen and under greenhouse, field, and nursery conditions, were effective in colonizing grapevine wounds or preventing and reducing vascular streaking caused by *P. chlamydospora* [1,30,34].

#### 3.1.1. Biocontrol Using *Trichoderma*

When hyphae of *P. chlamydospora* and those of *Trichoderma* species interacted, overgrowth of the pathogen, competition for nutrient, and direct antagonism were reported as mechanisms in the literature. For instance, the endophyte strains of *T. atroviride* (ATCC 74058) and *T. harzianum* (ATCC 26799) were able to outcompete or utilize more carbon and nitrogen sources than *P. chlamydospora*, significantly reducing the growth of the pathogen by 90% [35]. Kotze et al. (2011) also reported that commercial strains of *T. harzianum* and *T. atroviride*, as well as two *T. atroviride* (USPP-T1 and USPP-T2) strains isolated from the grapevine, overgrew *P. chlamydospora* after stopping its growth. The same result was obtained by Silva-Valderrama et al. (2021), who reported that the endophytic antagonist *Trichoderma* sp. Altair completely overgrew *P. chlamydospora* on day 21 of coincubation *in vitro*. Recently, Spasova et al. (2022) created an ecofriendly hybrid nanomaterial made of poly(l-lactic acid) fibers (PLLA) coated with chitosan and *T. asperellum* spores. Due to its good mechanical properties, this nanomaterial ensured the viability of the *T. asperellum* spores. When tested *in vitro*, it significantly suppressed the growth of *P. chlamydospora* [72].

*Trichoderma* spp. have been tested for their ability to protect grapevine pruning wounds in experiments in nurseries and in the field. For instance, the isolate USPP-T1 reduced the incidence of *P. chlamydospora* by 77% 8 months after inoculation under field conditions [34]. Mycoparasitism was suggested as the mechanism of action. It was reported that the application of *T. harzianum* at rooting in an organic nursery reduced the rate of *P. chlamydospora* infection over time [73]. Regarding vine cuttings and pruning wounds, their protection against *P. chlamydospora* infection were evaluated in the nursery and in the field [74]. Cuttings were dipped in a *Trichoderma* suspension of *T. harzianum* T39 (Trichodex^®^) and *T. longibrachiatum* before or after callusing. In the case of pre-callusing dips, conflicting results were yielded for the 3 years of the study; however, in the post-callusing treatment, *Trichoderma* spp. led to a significant reduction in necrosis length, caused by *P. chlamydospora* inoculated into the rootstock. As *Trichoderma* spp. and *P. chlamydospora* were never in contact, Marco and coworkers (2004) suggested that an enhancement of the grapevine defenses was responsible for the protective effect. In the same study, pruning wounds were also protected against *P. chlamydospora* infection under field conditions, with the two biocontrol agents *T. harzianum* T39 and *T. longibrachiatum* being reisolated 2 months after spraying [74].

As for rootstock, soaking of the planting material naturally infected by *P. chlamydospora* in *Trichoderma* formulation was applied in South African nurseries, reducing the incidence of the pathogen in the rootstock cuttings [75]. Martínez-Diz et al. (2021) also dipped the roots and the basal part of the plant in *Trichoderma koningii* TK7 suspension for 24 h before planting, and the incidence of *P. chlamydospora* infection in the field on young grafted Spanish Tempranillo cultivar was significantly reduced.

#### 3.1.2. Biocontrol Using Other Fungal Genera

In addition to *Trichoderma*, other fungal genera have been used in the literature to control *P. chlamydospora*, including endophyte isolates of *Epicoccum* spp. taken from the woody tissue of the cultivar Touriga Nacional grown a vineyard in Portugal. *E. layuense* E24 was the most efficient strain *in vitro* as it reduced *P. chlamydospora* growth by 79.9% [76]. Its mode of action was probably via competition for space as *E. layuense* E24 grew faster than the pathogen, or via chemical interaction as it produced diffusible pigments on the medium [76]. *E. layuense* E24 was, therefore, tested under greenhouse conditions on rooted cuttings of two grapevine cultivars against *P. chlamydospora*. *E. layuense* E24 colonized the wood without impairing plant growth or inducing the appearance of symptoms in leaves or wood. It reduced the frequency of the pathogen re-isolation and the brown wood streaking length in Cabernet Sauvignon and Touriga Nacional by 67.5% and 73.8%, respectively [76].

Other *in vitro* tests were conducted with *C. rosea* and *Lecanicillium* spp. strains. Rhizospheric and endophytic strains of *C. rosea* were cocultured with *P. chlamydospora in vitro*, and *C. rosea* almost completely overgrew (99.9%) *P. chlamydospora* by 21 days, presumably inhibiting pathogen growth through antibiosis and mycoparasitism [30]. *L. lecanii* (ATCC 46578) reduced the growth of *P. chlamydospora* by 50% *in vitro* and was able to outcompete it for carbon and nitrogen resources [35]. Five endophytic strains of *C. rosea* isolated from grapevine cv. Cabernet Sauvignon were effective in inhibiting *P. chlamydospora* growth *in vitro*; this inhibition was observed before *C. rosea* physical contact with the pathogen, which led the authors to suggest that the pathogen growth inhibition was due to the production of antibiotic compounds by *C. rosea* [40].

A strain (i.e., F2) of *Fusarium oxysporum* was isolated from a suppressive compost amendment [77]. It reduced the growth of *P. chlamydospora* by 43% and its sporulation by 90% *in vitro* at 28 days; nonetheless, no reduction in the discoloration length inside the trunk was observed, even though the DNA amount of *P. chlamydospora* was reduced by 82% in the presence of this antagonist [78]. The F2-treated grapevines also harbored higher lignin levels. The F2 strain was re-isolated 90 days after treatment, suggesting that it probably colonized the xylem tissues [78].

*In planta* under greenhouse conditions, the endophytic isolate *C. rosea* 19B/1 was assessed against *P. chlamydospora* on 1 year old grapevine cuttings grown for 90 days in greenhouse conditions in soil with 10^4^/g conidia of *C. rosea* 19/B1. The results showed that the length of the necrotic lesions caused by *P. chlamydospora* significantly decreased in the case of cuttings planted in *C. rosea*-amended soil [40]. In this case, when the pathogen growth was inhibited without any direct contact with *C. rosea*, the authors suggested two possible mechanisms of action: the first by inhibiting the pathogen growth by antibiotic compounds secreted in the soil or in the vascular tissues at the base of the cuttings, and then transported by the sap to the point of *P. chlamydospora* inoculation; the second by triggering the plant defense mechanism [40].

#### 3.1.3. Biocontrol Using Oomycetes

As for oomycetes, experiments were conducted using *P. oligandrum*. This oomycete was reported to protect grapevine against Esca pathogens, by inducing the plant defense of Cabernet Sauvignon cuttings in controlled greenhouse conditions [53,79,80]. Yacoub et al. (2016) and Daraignes et al. (2018) observed that the application of this oomycete at the root level reduced *P. chlamydospora* necroses in the stem. As there was no contact between the two microorganisms, with *P. oligandrum* applied in the soil surrounding the roots and *P. chlamydospora* present in the aerial parts, Yacoub et al. (2016) pointed out an enhancement of plant defense responses subsequent to pathogen infection. Six genes involved in various plant defense pathways, including PR proteins, phenylpropanoid pathways, oxylipin, and oxidoreduction systems, were more significantly expressed in the presence of the oomycete [80]. This *P. oligandrum* induced plant systemic resistance and was associated with the promotion of jasmonic/ethylene signaling pathways [79]. The effects of the combination of *P. oligandrum* with *Pantoea agglomerans* or *Bacillus pumilus* in young grafted grapevines under greenhouse conditions against *P. chlamydospora* were not significantly different from the single bacterial strain applications; hence, no synergic effect between these MBCAs took place in protecting against this pathogen [53]. Under field conditions, the strain *P. oligandrum* Po37 significantly reduced *P. chlamydospora* infection on young grafted Spanish Tempranillo cultivar [81].

#### 3.1.4. Biocontrol Using Bacteria

Although strains from the *Bacillus* genus have been extensively used, strains from other genera, i.e., *Acinetobacter*, *Brevibacillus*, *Curtobacterium*, *Enterobacter*, *Paenibacillus*, and *Pantoea*, have also been tested for their ability to control *P. chlamydospora*.

Bacteria have been isolated from grapevine and tested for their efficacy in controlling *P. chlamydospora*. Some experiments showed that, when *B. subtilis* interacted directly with *P. chlamydospora* hyphae, swelling and malformation on the pathogen’s hyphae were observed, suggesting antibiosis as the most likely mechanism of action [34]. Alfonzo et al. (2009), showed that heat-stable metabolites of *B. subtilis* AG1 inhibited *P. chlamydospora* growth [64]. In the vineyard, when *B. subtilis* was applied on the surface of fresh pruning wounds in a South African vineyard, there was a decrease in *P. chlamydospora* incidence 8 months after infection [34].

Andreolli et al. (2016) isolated endophytic bacteria from 3 and 15 year old grapevine stems of *V. vinifera* cultivar Corvina. They reported that *Bacillus* sp. 3R1 and 3R4, which clustered with the species *B. methylotrophicus*, had a promising antagonist effect on *P. chlamydospora in vitro* [49], and they were able to colonize the xylem tissue of grapevine. The endophytic bacterial strain AG1 of *B. amyloliquefaciens* [65] produced heat-stable metabolites and inhibited mycelial growth *P. chlamydospora in vitro* [64].

Haidar et al. (2016) assessed the antagonist activity of 46 bacterial isolates obtained from grapevine wood or grape berries, sampled from a Bordeaux vineyard (France). Eight among the 46 significantly reduced the necrosis length produced by *P. chlamydospora* on rooted cuttings of a Cabernet Sauvignon cultivar under greenhouse conditions. *Bacillus pumilus* (S32) *and Paenibacillus* sp. (S19) were the most efficient ones [82]. These two bacterial isolates exhibited a direct action on *P. chlamydospora*, by producing volatile compounds and a diffusible antibiotic substance *in vitro* that inhibited the pathogen growth; moreover, when inoculated alone, they induced the expression of defense-related genes on the grapevine 4 days after their application. This effect was only maintained in the leaves of plants treated with *B. pumilus* (S32) and *P. chlamydospora*, 15 days after pathogen inoculation. Haidar et al. (2016) suggested that *B. pumilus* (S32) induced systemic resistance in grapevine.

In addition to the *Bacillus* and *Paenibacillus* genera, other bacteria isolated from grapevine have been tested against *P. chlamydospora*. The same research group of Haidar et al. (2016) evaluated the biocontrol capacity of *Enterobacter* sp. (S24), *Paenibacillus* sp. (S18), *B. reuszeri* (S28, S31, and S27), *Bacillus* sp. (S34), *P. illinoisensis* (S13), *Pantoea agglomerans* (S1 and S3), and *Bacillus firmus* (S41) isolated from a Bordeaux vineyard against *P. chlamydospora*. The result of the bioassay under greenhouse conditions on rooted cuttings of Cabernet Sauvignon cultivar (INRAE, Bordeaux, France) showed that all bacterial strains significantly reduced the length of the internal necrosis after the artificial co-inoculation of the stem cuttings by *P. chlamydospora;* the strains *B. reuszeri* (S27) and *Enterobacter* sp. (S24) were less effective. In each case, Haidar et al. (2016) provided evidence that the application method, i.e., co-inoculation was prevented in the wood, and soil inoculation did not affect the efficiency of these potential MBCAs. Some authors reported the inefficacy *in vitro* of bacterial strains, such as one of *Acinetobacter radioresistens* and one of *Curtobacterium* sp., against *P. chlamydospora* [1].

Not all bacteria tested were isolated from grapevine; bacterial strains from the *Pseudomonas* genus, i.e., *Pseudomonas protegens* strain MP12, obtained from a forest soil sample, and *P. protegens* strain DSM 19095T, significantly inhibited *P. chlamydospora* growth *in vitro* [50]. A mixture of two strains, *Pseudomonas fluorescens* and *Bacillus atrophaeus* (Stilo Cruzial^®^), was tested under field conditions on 2 and 3 year old Tempranillo cultivar in Spain, against *P. chlamydospora* Petri disease, by soaking the roots and the basal part of the plant in the two bacterial suspensions for 24 h before planting. However, this process was inefficient against *P. chlamydospora* [81].

In a recent study, *Paenibacillus alvei* K165, isolated from the root tips of tomato plants grown in solarized soils, was tested for its ability to control *P. chlamydospora* on a growth medium simulating the xylem environment [83]. *In planta*, Gkikas et al. (2021) showed that the growth and the sporulation of *P. chlamydospora* were not inhibited by this strain; however, when tested on potted grapevines of cultivar Soultanina, the strain K165 reduced the endophytic DNA amount of *P. chlamydospora* by 90%, and the wood discoloration length in K165-treated vines was significantly reduced [78].

#### 3.1.5. Biocontrol Using Actinobacteria

Álvarez-Pérez et al. (2017), evaluated the effectiveness of two actinobacterial strains isolated from the root system of a 1 year old grafted *V. vinifera* cultivar Tempranillo. They were an endospheric strain and a rhizospheric strain, *Streptomyces* sp. E1 and *Streptomyces* sp. R4, respectively. In three experimental open-root field nurseries of young grafted *V. vinifera* cv. Tempranillo plants, there was a significant reduction in the infection rates at the lower end of the rootstock by *P. chlamydospora*, in the context of Petri disease [84]. However, Martínez-Diz et al. (2021) showed that the antagonistic effect of the strains *Streptomyces* sp. E1 and R4 put together was very low against Petri disease [81]. This points out the complexity and the variability of plant protection induced by MBCAs.

Overall, most *Trichoderma* isolates have been reported to be effective *in vitro*, as well as under field conditions for some strains. Other fungal genera showed a good efficiency *in vitro* and/or *in planta.* The ability of the oomycete *P. oligandrum* and some actinobacteria isolates were also evaluated and gave promising results in the control of *P. chlamydospora*. Indeed, the growth of this pathogen was inhibited by numerous bacterial strains *in vitro*, as well as *in planta* under controlled conditions.

### 3.2. Biological Control of Phaeoacremonium minimum

To control the Esca-associated fungus *P. minimum*, several strains of fungi, bacteria, oomycete, and actinobacteria have been used in various assays.

#### 3.2.1. Biocontrol Using Fungi

*Chaetomium* spp., *Epicoccum* spp., *Lecanicillium lecanii*, and *Trichoderma* spp. were among the fungal strains evaluated [1,35,76,85].

Strains of *Trichoderma* spp. have been used extensively against *P. minimum*. In the experiment conducted by Kotze et al. (2011) *in vitro*, commercial *T. atroviride* and *T. harzianum* strains, alongside two *T. atroviride* strains (coded USPP-T1 and USPP-T2), were able to stop the growth of the pathogen, with some strains coiling or disintegrating the pathogen hyphae. The mode of action of *Trichoderma* spp. in stopping *P. minimum* has been extensively studied using phenotype microarrays [35], suggesting that they may compete on “nitrogen plus carbon” and “carbon” sources with this pathogen. Wallis (2021) proposed direct antagonism and competition for nutriment as the main mechanism of action. Nanomaterial made of poly(l-lactic acid) fibers (PLLA), in which *T. asperellum* spores were incorporated, significantly inhibited the growth of *P. minimum in vitro* [72].

An experiment conducted under semi-field conditions on potted vine plants in a protected environment also provided evidence of antifungal activity within the plants. Carro-Huerga et al. (2020) showed that the endophyte strain *Trichoderma* T154 colonized the xylem vessels, fibers, and parenchymatic tissues inside the wood up to 12 weeks after inoculation. They also showed a reduction in plant colonization by *P. minimum*. These authors observed that the antagonistic effect of this strain was related to mycoparasitism, mainly via the adhesion of spores to the pathogen hyphae and competition for a niche by colonizing the xylem vessels [86]. Under field conditions on young grafted grapevine cultivar Tempranillo, the strains *T. atroviride* SC1 and *T. koningii* TK7 significantly reduced *P. minimum* infection [81].

Some endophytic fungal strains have been used to protect grapevines against *P. minimum*. Three strains (*Epicoccum layuense, Epicoccum mezzettii* E17, and *Epicoccum layuense* E33 isolated from the wood of grapevines (cv. Touriga Nacional, Portugal), were able to inhibit *P. minimum in vitro*, while other strains isolated from the same vineyard were not effective against this pathogen [38]. Due to the fast-growing ability of these antagonists, along with the diffusion in the culture medium of pigments, Del Frari et al. (2019) suggested that antagonism was primarily due to competition for space and nutrients, as well as probably chemical interaction. Another fungus, *Chaetomium* sp., had a significant *in vitro* efficacy against *P. minimum* [1]. Geiger et al. (2022) studied the antagonist capacity of five endophytic strains of *C. rosea* isolated from the grapevine *in vitro* and demonstrated that none of these strains were effective in inhibiting *P. minimum* growth.

Under greenhouse conditions, the strain *E. layuense* E24, isolated from cane woody tissue, was tested *in vivo* on two young grapevine cultivars. The wood symptomatology caused by *P. minimum* was significantly reduced when interacting with *E. layuense* E24, as well as unevenly between cultivars, with the best reduction for Cabernet Sauvignon (82%) compared to Touriga Nacional cultivar (31.3%) [38].

#### 3.2.2. Biocontrol Using Oomycetes

A biocontrol oomycete, *P. oligandrum* (strain Po37), significantly reduced *P. minimum* infection associated with Petri disease on grafted young grapevine cultivar Tempranillo under field conditions in Spain [81].

#### 3.2.3. Biocontrol Using Bacteria and Actinobacteria

Bacterial strains have been tested against *P. minimum*. A number of these strains were isolated from grapevine. Among them, many belonged to the genus *Bacillus*. The antagonistic activity of *B. subtilis* was assessed *in vitro* and under field conditions against *P. minimum*. In a dual-culture assay, the growth of this pathogen was inhibited by the antagonistic bacteria isolate, and antibiosis was hypothesized as the mechanism of action, as suggested by the malformations and swelling seen on pathogen’s hyphae [34]. Two endophytic strains of *Bacillus* sp. isolated in Italy from a 3 year old *V. vinifera* cultivar Corvina caused significant *in vitro* inhibition of *P. minimum* growth [49]. Another endophytic strain of *Bacillus licheniformis* was isolated from *V. vinifera* cv. Glera and inhibited the growth of *P. minimum* in dual culture [87]. The antagonistic effect of the strain *B. subtilis* AG1 isolated from grapevine wood tissues affected by Esca and its heat-stable metabolites showed their efficacy against *P. minimum* [64]. In 2012 this strain was reclassified as *B. amyloliquefaciens* [65].

Combinations of *Bacillus* strain with other bacteria have also been proposed. In a recent study, *Pseudomonas fluorescens* plus *Bacillus atrophaeus*, mixed in the commercial product Stilo Cruzial^®^, were assessed on young (2 and 3 years old) Spanish vineyard cultivar Tempranillo. This biocontrol product was applied by soaking the root and the basal part of the plant in a suspension of the two bacterial strains 24 h before planting; however, no effect was observed against *P. minimum* [81].

Two strains of *Pseudomonas proteges* exhibited antagonistic activity *in vitro* against *P. minimum* [50]. The growth of *P. minimum in vitro* was highly inhibited by *S. plymuthica*, *B. velezensis*, and *P. chlororaphis* [51].

Other bacteria and actinobacteria were isolated from the wood tissue of symptomatic and asymptomatic grapevine of two cultivars Glera grafted onto SO4 rootstock and Sylvoz-trained 20 year old plants [88]. Among the 38 selected bacterial strains, 24 were clustered with the Actinobacteria branch, in addition to 13 with Rhizobiales and one with Pseudomonadales. Most of these strains were able to overgrow *P. minimum in vitro*, and three of them showed high biocontrol potential against this pathogen (one of the three strains was identified as *Micromonospora* sp.).

Other actinobacteria strains showed good efficiency in reducing the infection rates at the lower end of the rootstock on young grapevine in the field [84].

Overall, most *Trichoderma* spp. strains tested *in vitro* significantly reduced *P. minimum* growth. Only a few strains were evaluated *in planta*, some of which showed good potential in controlling this pathogen. Other fungal and bacterial genera showed promising results *in vitro* and/or *in planta* with a constant efficiency, whether *in vitro* or *in planta*. Regarding the oomycete *P. oligandrum*, only one study reported its ability to reduce grapevine infection by *P. minimum* in the field.

### 3.3. Biological Control of Fomitiporia mediterranea

*F. mediterranea* is mainly isolated from sectorial and central white rot necrotic tissues [89], and only a few experiments with MBCAs have been performed to control it.

#### 3.3.1. Biocontrol Using Fungi

Among the fungi, *Epicoccum* strains isolated from cane woody tissue of cultivar Touriga Nacional from a vineyard in Portugal were mainly used against *F. mediterranea*. *E. mezzettii* E17 and *E. layuense* E7 strains inhibited the growth of *F. mediterranea* after 14 days of growth on PDA medium [38]. *E. mezzettii* E17 overgrew this pathogen, suggesting that the antagonistic activity is primarily due to competition for space and nutrients. On the other hand, the strains assigned to *E. layuense* species inhibited *F. mediterranea* growth without physical contact between the colonies and with pigments observed on the culture medium, suggesting the production of chemical inhibiting compounds. The most efficient *E. layuense* strain reduced the fungal colony size by 71.8%. Microscopic observation of the confronting zone revealed that *F. mediterranea* responded by entangling their hyphae, forming hyphal strands [38].

#### 3.3.2. Biocontrol Using Bacteria

Several biocontrol bacteria; strains have been isolated from various wood tissues of symptomatic *V. vinifera* (21 year old Sauvignon Blanc cultivar) from the Bordeaux region (France). Haidar et al. (2021) tested the antagonistic activity of 59 of these bacterial strains against *F. mediterranea* strain Ph CO 36, *in vitro*. Thirty-five strains out of 59 effectively inhibited pathogen growth in a dual-culture assay, while 99% of them inhibited its growth by secreting volatile compounds. The strains *Pseudomonas* sp. S45, *Stenotrophomonas* sp. S180, and *Novosphingobium* sp. S112 were the most effective against *F. mediterranea in vitro* (more than 50% inhibition in confrontation and volatile tests. Not all these bacteria had a deleterious effect on the pathogen; an additional five strains, *Enterobacter* sp. S11, *Paenibacillus* sp. S150, *Weeksellaceae* S259, *Paenibacillus* sp. S270, and *Bacillus* sp. S5, even promoted the growth of *F. mediterranea* in dual culture [71].

Overall, the biocontrol of *F. mediterranea* started in 2019, and strains of the genera *Epicoccum* have only been tested *in vitro* against this pathogen. Some of these strains showed a high efficiency in controlling *F. mediterranea in vitro*. So far, there are no *in planta* reports controlling this pathogen; thus, further studies aimed at selecting MBCAs against this pathogen are required.

## 4. Biocontrol of Eutypa Dieback

Eutypa dieback is a severe disease of vineyards that has been known for over 60 years, mainly caused by the Ascomycete *Eutypa lata* [9,90]. However, reports showed that some species of the family Diatrypaceae such as *Eutypa leptoplaca*, *Cryptovalsa ampelina*, and *Eutypella vitis* could also be involved in Eutypa dieback [9,21,91]. Regarding *E. lata*, ascospores are produced by perithecia on dead wood tissues after rain, before being released and dispersed by the wind. Through fresh pruning wounds, these ascospores penetrate and germinate in the xylem vessels, and the mycelium *E. lata* slowly colonizes the woody tissues [92,93].

In Eutypa dieback, the mechanisms involved in foliar symptoms and wood necrosis development are not well understood. Mahoney et al. (2003) reported that they may be caused by several *E. lata* metabolites, such as eutypine, which is the most phytotoxic [9,94,95]. *E. lata* synthesizes eutypine in wood tissue, which is likely transported by the ascending sap flow to the herbaceous parts of the vine, before penetrating the grapevine cells via passive diffusion, and then accumulating in the cytoplasm due to ion-trapping mechanism related to the ionization state of the molecule [96]. When this phytotoxin targets the mitochondria, it causes inhibition and uncoupling of mitochondrial oxidative phosphorylation [9,96]. The enzymatic reduction of eutypine gives its corresponding alcohol, eutypinol, which is not toxic to grapevine, suggesting this as a detoxification pathway for eutypine [9]. In addition to eutypine, other phytotoxic compounds are thought to be involved in foliar symptoms [3,97,98]. For instance, Andolfi et al. (2011) reported that *E. lata* produced related secondary metabolites, mainly acetylenic phenols, along with some low-molecular-weight metabolites involved in the chelator-mediated Fenton (CMF) reactions that generate highly damaging reactive hydroxyl radicals, likely inducing necrosis on grapevine wood tissue [3].

### 4.1. Biological Control of Eutypa lata

To control *E. lata in vitro* and *in planta*, very different results in terms of effectiveness have been obtained with potential biocontrol fungal, bacterial, and actinobacterial strains.

The following species have been tested: *Trichoderma atroviride*, *T. guizhouense*, *T. harzianum*, *T. koningiopsis*, *T. longibrachiatum*, *T. paraviridescens*, *T. spirale*, *T. afroharzianum*; *T. simmonsii*, *Lecanicillium lecanii*, *Fusarium lateritium*, *Rhodotorula rubra* (yeast), *Candida famata* (yeast), *Penicillium* sp., *Alternaria alternata*, and *Cladosporium herbarum* [31,33,34,35,42,99,100,101].

#### 4.1.1. Biocontrol Using *Trichoderma*

*In vitro*, the mode of action of *Trichoderma* species against *E. lata* hyphae, which is strain-dependent, has been extensively studied, involving either production of inhibiting metabolites or mycoparasitism. For instance, John et al. (2004) showed that the mycelial growth of *E. lata* was inhibited by volatile and nonvolatile metabolites produced by three strains of *T. harzianum* (AG1, AG2, and AG3). Similarly, Kotze et al. (2011) observed these two modes of actions with *T. harzianum* and *T. atroviride* strains obtained from biocontrol commercials products, i.e., Biotricho^®^, Vinevax^®^, and Eco 77^®^, as well as two *T. atroviride* strains (USPP-T1 and USPP-T2) isolated from South African grapevine. At the microscopic level, the hyphae of *T. atroviride* AG5 and *T. harzianum* Eco 77 coiled around those of the pathogen, indicating mycoparasitism activity, but there was also a clear inhibition zone between the cultures of USPP-T1 and USPP-T2 strains and those of *E. lata*, suggesting that antibiosis occurred, as shown by the production of volatile and nonvolatile inhibiting metabolites. Kovács et al. (2021) reported that *T. afroharzianum* TR04 and *T. simmonsii* TR05 strains, obtained from grapevine cordon wood, displayed mycoparasitism against *E. lata*, as seen by the hyphal coiling and penetration of the pathogen’s hyphae.

Screening of strains from various *Trichoderma* species obtained from different ecosystems, according to their antagonistic ability against *E. lata*, was conducted by Úrbez-Torres et al. (2019). Regarding *T. atroviride*, the most efficient strain was the isolate *T. atroviride* PARC1018 that inhibited the mycelial growth of this pathogen by 68.2%, while three other strains from this species caused a slight inhibition (less than 50%) of *E. lata*. Regarding the other *Trichoderma* species, *T. harzianum* PARC1019, *Trichoderma* sp. PARC1020, *T. koningiopsis* PARC1024, and four strains of *T. guizhouense*, obtained from Italian *P. persica*, as well as the strain *T. paratroviride* PARC1012, significantly reduced the *E. lata* mycelium growth by more than 50%, whereas, for others strains, i.e., *T. harzianum* PARC1013, *T. longibrachiatum, T. paraviridescens*, and *T. spirale* obtained from *P. persica*, this mycelial growth reduction was lower [33]. *E. lata* growth was significantly reduced by about 70% by the endophytic strains *T. atroviride* (ATCC 74058) and *T. harzianum* (ATCC 26799), which outcompeted or utilized more carbon and nitrogen sources than *E. lata*, suggesting competition for nutrients as the mechanism of action [35].

*Trichoderma* spp. were mainly implemented in field trials to protect grapevine pruning wounds. In South Africa, Halleen et al. (2010) showed that, on Cabernet Sauvignon vineyards (8 and 10 years old) artificially infected by *E. lata*, protection of pruning wounds was higher with chemical products (benomyl and flusilazole) than with *Trichoderma* biocontrol products (Trichoseal-Spray, Eco 77^®^, and Biotricho^®^). On four other South African young vineyards (5–9 years old) naturally infected with *E. lata*, two of them with grape cultivars Cabernet Sauvignon and Sauvignon Blanc, and two others with table grape cultivars, Red Globe and Bonheur, *Trichoderma*-based biocontrol products, Vinevax^®^ and Eco 77^®^, reduced the natural infection of pruning wounds, but their efficiency varied according to the season and the cultivars [101].

Another treatment consisted of brushing the wounds with spores of *Trichoderma* spp. [100]. Indeed, in a glasshouse assay, *T. harzianum* AG1 prepared in sterile distilled water or three commercial formulations, i.e., Trichoseal, Trichoseal spray, and Vinevax, applied by brushing within the first hour of pruning, reduced the recovery of *E. lata* when the ascospores of the pathogen were inoculated 2 or 7 days after pruning [100]. The same reduction in E. *lata* recovery was observed under field conditions in a South Australian healthy vineyard (16 year old Cabernet Sauvignon cultivar) when spores of *T. harzianum* or Vinevax were applied by brushing on fresh pruning wounds 1 or 14 days before the inoculation of *E. lata* ascospores [100].

#### 4.1.2. Biocontrol Using Other Fungi

*In vitro*, according to Wallis (2021), *L. lecanii* (ATCC 46578) did not significantly reduce *E. lata* growth, but outcompeted or utilized more carbon and nitrogen sources than the pathogen. Blundell et al. (2021) investigated the biocontrol ability of two strains of *A. pullulans* isolated from cane tissue and the sap of Chenin Blanc cultivar grapevines in California. These two strains, UCD 8344 and UCD 8189, were inefficient in a dual-culture assay against *E. lata*. Five endophytic *C. rosea* strains obtained from grapevine were effective in inhibiting *E. lata* growth *in vitro*, without any direct contact [40].

John et al. (2005) showed that the treatment of pruning wounds by spores of the saprophyte *Fusarium lateritium* reduced the recovery of *E. lata* when the antagonist was applied at least 1 day before the pathogen in a glasshouse assay, as well as on 16 year old healthy vineyards of the cultivar Cabernet Sauvignon in South Australian.

Munkvold and Marios (1993) evaluated the ability of 348 fungal strains isolated from grapevine pruning wounds of cultivar Chenin Blanc to inhibit *E. lata in vitro* on excised grapevine stems. Among these isolates, 49% did not reduce *E. lata* infection on the non-autoclaved wood, and only 1% completely reduced the infection of the wood stems by this pathogen. These authors conducted two field bioassays on 21 year old Thompson seedless grapevines and showed that the two strains of *F. lateritium* and *Cladosporium herbarum* significantly reduced the infection of pruning wounds by *E. lata*, when applied using a brush immediately after pruning. Depending on the bioassay, this reduction was equal to or greater than with the fungicide treatment, i.e., with benomyl. Regarding the modes of action of the two strains, *F. lateritium* certainly acted via antibiosis, as it produced a diffusible metabolite that inhibited *E. lata in vitro*; regarding *C. herbarum*, as it has a high rate of hydrophobic conidia sporulation, competition for space through the colonization of pruning wounds by these conidia was probably its main antagonistic mechanism [102].

Under greenhouse conditions, Geiger et al. (2022) conducted an *in planta* study on 1 year old grapevine cuttings grown for 90 days in greenhouse in soil with 10^4^/g conidia of *C. rosea* 19/B1. They showed that *C. rosea* 19/B1 significantly reduced the length of the necrotic lesions caused by *E. lata*. Because *C. rosea* inhibited the pathogen growth without any direct contact with it, Geiger et al. (2022) suggested that the mechanism of action of *C. rosea* was antibiosis or induction of the plant defense mechanism.

#### 4.1.3. Biocontrol Using Bacteria and Actinobacteria

Strains from the following bacterial species have displayed some potential to act as biocontrol agents against *E. lata*,: *Bacillus cereus*, *Bacillus megaterium*, *B. subtilis*, *Bacillus thuringiensis*, *B. velezensis*, *Erwinia herbicola* (syn *Pantoea agglomerans*), *Micrococcus kristianae*, *Pseudomonas* sp., *Pseudomonas aeruginosa*, *Pseudomonas fluorescens*, *Serratia plymuthica*, and *Stenotrophomonas maltophilia* [34,42,103,104].

Among the *Bacillus* species, *B. subtilis* has been extensively used *in vitro* and under field conditions.

*In vitro*, *B. subtilis* strains were isolated from the wood of arms of Chenin Blanc grapevine expressing Eutypa dieback symptoms or from compost soil. The strain coming from grapevine inhibited *in vitro* the mycelial growth of *E. lata* by 91.4% and the ascospore germination by 100% [104]. In dual culture, *B. subtilis* caused malformations [34,104] and swelling in *E. lata* hyphae [34]. As Ferreira et al. (1991) identified at least two antibiotic substances that were responsible for the inhibition of *E. lata* mycelial growth and ascospore germination, antibiosis was suggested as the mode of action of this *B. subtilis* strain [34,104]. Schmidt et al. (2001) also hypothesized antibiosis as the mode of action when they used the liquid cultures of two *B. subtilis* strains isolated from boiled compost soil. They caused at least 50% suppression of *E. lata* mycelial growth over 2 weeks on autoclaved discs of perennial grape wood cultivar Müller Thurgau (over 10 years old).

From 10 year old cultivar Chenin Blanc grapevines in California (USA), Blundell et al. (2021) isolated one bacterial strain from the sap and another one from the cane pith of *V. vinifera*, which were identified as closely related to *B. velezensis*. These two strains inhibited *E. lata* mycelial growth *in vitro* in both dual-culture and volatile organic compound assays. In the dual-culture assay, an inhibition zone was observed between *E. lata* and *Bacillus* isolate UCD 8347, which also significantly inhibited the growth of *E. lata* in the volatile assay. Hence, the authors suggested that an antibiotic substance was produced by this isolate.

Under field conditions in South Africa, a *B. subtilis* strain isolated from grapevine wood was used to control *E. lata* infection on different cultivars of young and mature vineyards [34,104]. On a 4 year old Riesling vineyard, after pruning of grapevines, *B. subtilis* suspension and its antibiotic extract were sprayed directly on the pruning wound surface, while *E. lata* was inoculated 4 h after [104]. Nine months later, it was shown that the pathogenic infection was significantly suppressed by the bacterial suspension (100%). However, the antibiotic substance was ineffective in protecting pruning wounds from *E. lata* infection [104]. Kotze et al. (2011) showed that, when this bacterial strain was sprayed on fresh pruning wounds of grapevines from 10 year old Merlot and 18 year old Chenin Blanc vineyards, with *E. lata* was inoculated 7 days afterward, *E. lata* incidence was lower after 8 months compared to the control. Nevertheless, in another experiment realized in South Africa in 8 and 10 year old Cabernet Sauvignon vineyards, no efficiency was observed in protecting pruning wounds when using a suspension of bacterial strains against the artificial infection of *E. lata* 24 h after spraying [101]. These variations in efficiency were probably due to the difference in grapevine ages and cultivars used in each study. The spraying methods, i.e., covering or not the pruning wounds after treatment, and the interval of time between *B. subtilis* and *E. lata* inoculations also presumably influence the success or the failure of the biocontrol protection.

In addition to *B. subtilis*, other bacterial species/genera have been tested against *E. lata*. Schmidt et al. (2001) reported that *B. cereus*, *B. subtilis*, *B. thuringiensis*, *Pseudomonas aeruginosa*, eight strains of *Pseudomonas fluorescens* and one of *Stenotrophomonas maltophilia*, and an unidentified isolate belonging to the Enterobacteriaceae family inhibited *E. lata* growth in a coculture assay. However, none of these strains showed efficacy in controlling this fungal pathogen on discs of grapevine wood. In the same assay, an *Erwinia herbicola* strain (reclassified as *P. agglomerans*) isolated from the rhizosphere of a Gramineae species in west Germany [105], displayed antifungal activity against *E. lata in vitro*, as well as on wooden discs [103]. Its culture filtrate contained siderophores and antifungal molecules that inhibited *E. lata* growth on the wood discs; therefore, Schmidt et al. (2001) suggested that its mechanism of action was probably due to both antibiosis and a competitive effect under iron-limiting conditions. The same authors tested the potential of 104 Actinomycetes isolates *in vitro* against *E. lata* both in dual culture and on discs of grapevine wood assay. Seventeen isolates identified as *Streptomyces* sp. inhibited *E. lata* growth; one of these unidentified Actinomycetes isolates (A123) showed the highest degree of *E. lata* inhibition on wood, ranging from 70% to 100% over a 4 week period. Eighty percent of the unidentified Actinomycetes isolates inhibited *E. lata* in dual culture, but only 11% were efficient in the antagonism assay carried out on grapevine wood discs [103]. Recently, it was reported that 30 endophytic actinobacteria isolates obtained from grapevine were able to inhibit or moderately highly the growth of *E. lata in vitro* [54].

Munkvold and Marios (1993) showed that 60% of the 391 bacteria isolated from pruning wounds of the cultivar Chenin Blanc reduced the infection by *E. lata* on grapevine autoclaved stems, and only 2% (20 strains) were able to inhibit this pathogen. However, under field conditions, the strains *B. megaterium*, *M. kristianae*, and *P. fluorescens* were inefficient in protecting pruning wounds from *E. lata* infection, and the colonization of pruning wounds by the pathogen was even enhanced in the presence of a *B. megaterium* strain. Munkvold and Marios (1993) suggested that the high dose of ascospore inoculum of *E. lata* used for the field experiment reduced the efficacy of the strains tested [102].

Overall, to biocontrol *E. lata*, many strains of *Trichoderma* have been evaluated *in vitro* and *in planta* over the previous years. Some of them gave very good results *in vitro* and under field conditions, but the efficiency of MBCAs depends on factors such as the season of application and the grapevine cultivars. This was observed with some *Trichoderma*-based products evaluated in the vineyards against *E. lata*. Other fungal strains such as *C. rosea*, *Fusarium lateritium*, and *L. lecanii* effectively controlled *E. lata in vitro* and/or *in planta*. Bacterial isolates were also evaluated, but they showed variable agreement between the *in vitro* results and the results obtained in the field.

## 5. Biological Control with Currently Commercialized Products

As reported above, the grapevine’s defense responses are sometimes not sufficient to cope with the development of GTD fungal pathogens. However, no highly efficient treatment currently exists to prevent, protect, or even limit the progression of these diseases. Only a few products are registered to reduce Esca foliar symptoms: a product based on a foliar fertilizer mixture of calcium, magnesium, and seaweed (Algescar^®^, Natural Development Group, Castelmaggiore, Bologne, Italy) [2,19,106,107], and a few *Trichoderma*-based products registered in some countries. Furthermore, no commercial bacterial biocontrol products are registered against GTD pathogens [52]. In this section, we review products based on *Trichoderma* spp. strains to biocontrol GTD fungi.

Vintec^®^ is a fungicide based on 2 × 10^10^ CFU/g of *Trichoderma atroviride* strain SC1 spores [55], in the form of dispersible granules, which is applied by spraying. The strain *T. atroviride* SC1 is approved for use under EC 1107/2009 in several European countries as a fungicide. It is widely used to control various fungal pathogens involved in grapevine wood diseases such as *P. chlamydospora*, *P. minimum*, *D. seriata*, *Botryosphaeria ribis*, *E. lata*, and *Eutypa armenicae*, as well as for crop protection against gray mold (Table 1). *T. atroviride* SC1 was isolated from decayed hazelnut wood and selected as an MBCA for its high colonization ability and its good lignocellulolytic capacity. In addition, this strain can use mannose and galactose as carbon sources, which are the main components of the hemicellulose of softwood [108]. *T. atroviride* SC1 has antagonistic activity against several plant pathogens [109]; it is a fast-growing fungus that has no negative effects on plants but enhances plant growth by promoting nutrient assimilation, in addition to quickly colonizing the dead wood [110]. In nurseries, *T. atroviride* SC1 was more efficient when applied at hydration stages to control *P. minimum* and *P. chlamydospora* infections [111]. According to the producer’s recommendations, Vintec^®^ should be applied when the environmental temperature is equal to or higher than 10 °C for a minimum of 5 h the day of the field application [55]. Recently, Martínez-Diz et al. (2021) reported that soaking the roots and the basal part of the *V. vinifera* Tempranillo cultivar in *T. atroviride* SC1 suspension for 24 h before planting reduced the incidence of certain GTD fungi [81]. *T. atroviride* SC1 was very effective in preventing *P. minimum* and *P. chlamydospora* infection on grapevine pruning wounds in the field, as well as during the grafting process in nursery [111,112]. Berbegal et al. (2020) showed that this strain could reduce infections caused by some GTD pathogens when new vineyards were planted.

Chervin et al. (2022) determined that Vintec^®^ significantly reduced the wood colonization by *P. chlamydospora* and *P. minimum* on 1 year old canes of *V. vinifera* cv. Cabernet Sauvignon, planted in pots. By conducting metabolomic studies, these authors showed that the application of Vintec^®^ alone induced a weak metabolomic response that was not sufficient to stimulate plant defense mechanisms. Nevertheless, the application of Vintec^®^ with the pathogens attenuated the virulence, since some *P. minimum* and *P. chlamydospora* metabolites were highly produced in the control condition but less produced in the presence of Vintec^®^ [113]. This product also had an effect on the plant by priming its defense mechanisms. It seems that Vintec^®^ increased plant response with a stimulation of the phenylpropanoid pathway with increasing amounts of stilbenoid pterostilbene, as well as an increase in flavonoids. This allowed the authors to suggest a mechanism of action based on competition and the stimulation of plant defense mechanisms [113]. By performing a transcriptomic analysis, Romeo-Oliván et al. (2022) revealed that *T. atroviride* SC1 (Vintec^®^) enhanced modifications in the gene response to GTD, both alone and during *P. minimum* and *P. chlamydospora* infection. During infection by these pathogens, Vintec^®^ promoted the expression of genes related to the biosynthesis of stilbenes, phenols, and flavonoids, which are metabolites known for their antifungal properties. It also modulated the expression of some genes involved in hormonal signaling, especially those involved in auxin signaling. Accordingly, the authors suggested that Vintec^®^ enhanced the primary defense response of the plant against Esca-associated pathogens [114].

Esquive^®^ is another biological control product containing spores of the species *T. atroviride;* the strain coded I-1237 was originally isolated from the soil (BPDB). This product is approved for use under EC 1107/2009 in Cyprus, France, Italy, Spain, and Portugal (BPDB), as well as in Australia New Zealand, South Africa, and Vietnam [115], to prevent the infection of grapevine pruning wounds by Esca and Botryosphaeria-associated pathogens, as well as *E. lata*, and for the control of root diseases and damping off in fruits and vegetables (Table 1). Esquive^®^ contains 10^8^ UFC/g of live *T. atroviride* I-1237 spore [55], in the form of wettable powder used on leaves or via brush application on pruning wounds. The strain I-1237 colonizes the wood after its penetration through pruning wounds and protects the grapevine from GTD pathogens via various mechanisms of action, including the inhibition of pathogenic fungal growth by competing for nutrients and mycoparasitism (Table 1). Mounier et al. (2016) showed that the application of Esquive^®^ on pruning wounds of mature grapevines for at least 2 years reduced plant mortality and leaf symptoms associated with Botryosphaeria dieback, Eutypa dieback, and Esca [116]. Martínez-Diz et al. (2021) evaluated the efficiency of Vintec^®^ and Esquive^®^ after their application on two mature Spanish commercial vineyards (37 and 29 years old) during two seasons (2018–2019 and 2019–2020), but their results showed a low efficacy of these products against *P. chlamydospora* and *D. seriata* in the two vineyards over 2 years [55].

Eco77 is a bioproduct approved for use to control Botrytis in zucchini, tomato, and roses, as well as Eutypa dieback in grape in South Africa, Kenya, and Zambia [115]. This product is available in the form of wettable powder that contains 2 × 10^9^ spores/g of *T. harzianum* B77, applied by spraying on pruning wounds. The protection conferred by this strain is due to its ability to colonize the pruning wound and to compete for space and nutrients, thus preventing *E. lata* infection. Kotze et al. (2011) found that this strain might also produce antifungal metabolites *in vitro*, suggesting antibiosis.

Blindar is a mixture of two *Trichoderma* spp. strains, i.e., *Trichoderma asperellum* ICC012 and *Trichoderma gamsii* ICC080, that was approved for use on grapevine in the form of wettable powder. This product contains 20 g/kg of each *Trichoderma* spp. Blindar protects pruning wounds through various mechanisms of action: colonization of the wounds with activity on GTD fungi via antibiosis and mycoparasitism, as well as growth inhibition by competing for nutrients in the invasion sites (see BPDB website for more details). It is applied on the grapevine by spraying at the beginning of the season because the germination and growth of *Trichoderma* spores require favorable temperatures, and because the grapevine bleeding sap is rich in sugars that favor spore development. The two *Trichoderma* strains act via mycoparasitism (BPDB): *T. asperellum* on *P. chlamydospora* at 15 °C and *T. gamsii* at 10 °C. These Blindar strains are available in other countries within other commercial formulations, named Cassat WP, Escalator, Bioten WP, or Remedier^®^ (Table 1). Remedier^®^ is, for instance, commercialized and used in Italy to reduce the incidence of Esca and grapevine mortality in affected vineyards by protecting wounds from new infections. Like Blindar, the Remedier^®^ product contains 20 g/kg of *T. asperellum* ICC012 and 20 g/kg of *T. gamsii* ICC080 [13,117]. After multiyear treatment, this product provided good results starting from the second or third year of application [1]. It was reported that the spraying of solutions containing this product for 7 years at the phenological stage of bleeding in three Italian vineyards reduced Esca symptoms by 22% [118]. Recently, Di Marco et al. (2022) demonstrated the effectiveness of the preventive application of these products under field conditions, early after pruning and yearly after planting [119].

Vinevax is another product that contains five strains of *T. atroviride* and is available in two forms, Vinevax Biodowel and Vinevax™ Pruning Wound Dressing (Table 1), which are approved and commercialized in Australia and New Zealand. The first one is in the form of slow-release wood dowels, applied directly in the trunk to prevent and treat Eutypa dieback (*E. lata*) and Botryosphaeria dieback (*Botryosphaeria stevenssi* syn. *Diplodia mutila*) by stimulating the systemic protective response of the plant. Vinevax is also used on orchard and ornamental trees [115]. The second one is a wettable powder applied by spraying on grapevine pruning wounds. Its protective effect against airborne *Eutypa* ascospores is due to its ability to durably colonize the pruning wounds. It is also used on orchard trees against wood decay [115].

## 6. Mechanisms of Action of MBCAs against GTDs

The lack of effective strategies to manage GTDs and the need for the ongoing development of biocontrol products have prompted scientists to evaluate the biocontrol potential of numerous microbial strains against GTD fungi. After the ban of sodium arsenite, fungi were used in initial studies on GTD pathogen biocontrol, and some *Trichoderma* strains were registered and used in viticulture, but these products are not intended to specifically manage GTDs. The two species *T. atroviride* and *T. harzianum* are frequently used to control at least one of the GTD pathogens, and they are known to act via several mechanisms of action, such as mycoparasitism and competition for space and/or nutrients (Table 2). Within species of the *Trichoderma* genus (Table 2), the latter mode of action, i.e., competition for space and/or nutrients, presumably plays an important role in controlling GTD fungi since most pathogens penetrate grapevines through pruning wounds [33].

In comparison to fungal strains, the number of bacterial strains tested is high, with strains belonging to *Pseudomonas* and *Bacillus* genus being the most tested against GTD fungi; however, no bacterial products are currently available on the market. The mechanism of action of these potential biocontrol bacteria has been less addressed in the scientific literature, but antibiosis and induction of grapevine resistance by priming the expression of defense-related genes (Table 2) are the two most commonly cited. Among all potential MBCAs, Actinobacteria are less studied for their antagonistic activity against the GTD pathogens.

The oomycete *P. oligandrum* naturally colonizes the grapevine root system and protects it via the induction of systemic acquired resistance against several GTD pathogens (Table 2).

Antibiosis, competition for nutrients and space, the production of siderophores and hydrolytic enzymes, parasitism, and the induction of systemic resistance are the mechanism of action exhibited by the various MBCAs assessed against GTD-associated fungi (Figure 1).

## 7. Factors Influencing Control Efficiency of the MBCAs

Variability and environmental factors, e.g., climate and soil type, have an influence on the efficacy of the biological control agents in the field [120]. In the case of GTD biocontrol, efficiency depends on the strains, as those belonging to the same species have various levels of efficiency toward the same pathogen. For instance, various strains of *T. harzianum* had different levels of efficiency against *E. lata in vitro* [33,34,99] and *in planta* [34,99]. The same observation was made by Silva-Valderrama et al. (2021) for various strains of the species *C. rosea* against *N. parvum*.

The same result was obtained with bacteria, whereby strains belonging to the same species displayed different levels of efficiency when assessed against the same pathogen. This was reported by Haidar et al. (2016) who conducted assays *in vivo* and observed clear differences in the biocontrol efficiency of bacterial strains belonging to the same species against *N. parvum* and *P. chlamydospora*. This dissimilarity also depends on the pathogens targeted: (i) at the species level, Úrbez-Torres et al. (2020) showed that the same strains of *Trichoderma* spp. had different behaviors against *N. parvum*, *D. seriata*, and *E. lata*, while the same observation was reported by Kotze et al. (2011) when a strain of *Trichoderma* spp. responded differently to interactions with *N. parvum*, *D. seriata*, *L. theobromae*, *P. chlamydospora*, *P. minimum*, *N. australe*, *E. lata*, and *P. viticola*; (ii) at the strain level, Mondello et al. (2019) showed that *F. proliferatum* was highly effective *in vitro* against the strain *N. parvum* “Sainte Victoire”, but less effective against two other strains of *N. parvum* tested under the same conditions.

This efficiency dependence on the pathogen/antagonist strains was also reported by John et al. (2004) when metabolite production and action were considered. For instance, *T. harzianum* AG2 volatile metabolites were the most effective in reducing the growth of *E. lata* 280, while those of *T. harzianum* AG3 strongly inhibited *E. lata* CS-Ba.1 [99]. Because the production of metabolites is strain-dependent, even within the same species [99], the difference in efficiency depends strongly on the mechanism of action of MBCAs and the response of the pathogens. For instance, in the study of Kotze et al. (2011), an inhibitory effect between *T. atroviride* AG8 and *L. theobromae* in a dual culture was observed. This assumption was supported by Kotze et al. (2011), who observed that some strains of *T. atroviride* displayed antibiosis against *D. seriata*, while other strains employed mycoparasitism toward the same strain of *D. seriata*.

Inconsistency has also been observed, with some MBCAs being efficient *in vitro* but less so in the field. It has been reported that their efficiency may sometimes depend on the mode of application of the MBCAs *in planta*. Actually, Haidar et al. (2021) demonstrated that the inhibitory activity of some bacterial strains against *N. parvum* was strongly affected by the mode of application used, but had no effect on the efficacy of *F. lateritium* against *E. lata* according to Munkvold and Marios (1993), using five potential biocontrol bacterial strains against *P. chlamydospora* [82].

Formulation, the time of application, the phenological stage of the grapevine, its age, and the cultivar may also affect the efficacy of MBCAs *in vivo*, as well as the origin of the MBCAs [1].

The strains isolated from wood are more adapted to the physical and chemical conditions of the grapevine wood [30]. For trials *in vitro*, the difference in efficiency may also depend on the culture medium used in the experiment, as the medium’s chemical composition may guide the metabolism of the bacterial and fungal MBCAs in one way or another [121,122]. Drawing a parallel with these examples, Bardin et al. (2015) signaled that this would likely occur for plant pathogens, especially when the biocontrol products have a single mode of action. Therefore, these authors suggested that the hypothesis of the “durability of biological control being higher than that of chemical control” may not always be justified. Consequently, more research studies are required to anticipate the integration of durability concerns in the screening procedure of new biocontrol agents [123]. To our knowledge, no resistance toward MBCAs by GTD fungi has been reported, which is probably due to both their infrequent use and the complexity of their mechanisms of action; however, this topic has not been studied enough.

## 8. General Discussion: Challenges and Prospects

The management of GTDs is difficult because (i) several pathogens are involved in the same disease, (ii) the synergy between microorganisms degrades the wood [71], and (iii) more than one trunk disease can sometimes occur in the same plant [2,42,124].

To control GTDs, the selection of MBCAs tolerant to pesticides and resistant to antimicrobials and toxins present in the environment may present a promising approach to enhance their persistence and efficiency in the field. For instance, it was reported by French et al. (2021) that the alternative use of MBCAs and conventional fungicides reduced the levels of synthetic inputs and the risk of fungicide resistance. Using this strategy, promising results were obtained with the strains *T. atroviride*, *T. harzianum*, and *F. lateritium* that are benzimidazole-resistant [1,99,125], as well as *Trichoderma* strains (mainly *T. afroharzianum* and *T. simmonsii*) that are myclobutanil-resistant, when assessed against GTD pathogens [31]. Another factor that has to be taken into account is the resistance of MBCAs to the various antimicrobials and toxins in the environment, because they may affect their persistence and biocontrol efficiency. For instance, Gkikas et al. (2021) evaluated the biocontrol potential of the strains *P. alvei* K165 rifampicin-resistant mutant and *F. oxysporum* F2 hygromycin B-resistant mutant against *P. chlamydospora*. Accordingly, the selection of MBCAs tolerant to pesticides and resistant to antimicrobials and toxins present in the environment may present a promising approach to enhance their persistence and efficiency in the field.

Another approach that is widely used in biocontrol to optimize impact is the development of products containing multiple microbial strains with different modes of action [126]. Currently, a product available on the market (i.e., Blindar) is based on two *Trichoderma* strains, i.e., *T. asperellum* ICC012 and *T. gamsii* ICC080, which act by antibiosis and competition for nutrients and space against GTD pathogens (Table 1). Recently, Di Marco et al. (2022) demonstrated that the preventive application of this product significantly reduced the expression of Esca symptoms under field conditions, and they showed that its potential also persisted in the environment, as they re-isolated the two *Trichoderma* strains after 7 months. A mixture of *Trichoderma* and *Gliocladium* was effective in the field against *Phaeoacremonium* spp. and *P. chlamydospora*, but this was not the case for a bacterial mixture of three strains of *Azospirillum* sp., *Bacillus* sp., and *Pseudomonas* sp. against these pathogens under the same conditions [1]. In another study, no synergetic effect between *P. oligandrum* and *P. agglomerans* or *B. pumilus* was observed under greenhouse conditions in controlling *P. chlamydospora* [53]. Martínez-Diz et al. (2021) demonstrated that the combination of two or more beneficial MBCAs promoted the prevention of black foot and Petri diseases in vineyards. Another relevant point, associated with the mixture of biocontrol agents, is that the combination of beneficial microorganisms with different modes of action reduces the probability of resistance development by the pathogens.

Improving the methods of strain selection is, therefore, crucial to optimize the efficiency/persistence of the MBCAs. Temperature plays an important role in this regard. This enables relevant pieces of information to choose the optimal time to apply the product in the field. Another relevant issue to be checked is the ability of some endophytic species efficient to control one pathogen, but potentially stimulate another disease. This observation was reported by Haidar et al. (2016) with the strain *Bacillus* sp. S43, which inhibited *Botrytis cinerea* infection but increased the symptoms caused by *N. parvum* when applied on grapevine cuttings. Hence, interest in conducting comparative screening bioassays is of the utmost importance.

## 9. Conclusions

To control GTDs, it is now becoming increasingly clear that no single effective control measure must be used, with disease management based on integrated strategies combining various control methods such as physical, chemical, biological control, cultural practices, and tolerant grapevine cultivars being on the rise. These integrated strategies also include other techniques aimed at limiting the propagation of the pathogens and the infection risk, mainly during the nursery process and upon the plantation/establishment of new vineyards [8,127].

Another important point is that integrated management strategies to manage GTDs respond to the societal demand for low-environmental-impact and ecofriendly strategies of plant protection. As for integrated management strategies, the use of MBCAs to develop durable and ecological products to manage this devastative disease is also on the rise. However, the selection of useful strains remains a major challenge, especially with regard to optimizing the efficiency and the persistence of the MBCAs in the field, whether they consist of a single microbial strain or a mixture of strains. The taxa of MBCAs also play a major role in the selection process, since some microbial species or genera are known to produce toxins or to be potential plant pathogens (e.g., *Fusarium* and *Erwinia*). In addition, the selection of strains able to grow on variable nutrient sources (mainly carbon and nitrogen) may express a double advantage: a high potential for competition with pathogens and a low cost of industrial production.

For the future, another key challenge will be to decipher the microbiome of grapevine, since pathogens responsible for pathogenicity interact with the plant and its microbiome. It is assumed that potential endophytic MBCAs isolated from grapevine are highly effective against GTD fungi, because they are adapted to the wood tissue environment and they share the same host as the pathogens [30,42]. For this reason, further studies aimed at understanding the microbial interactions in the wood of diseased and healthy grapevine would be a key point for selecting microbial strains able to fight GTD pathogens. Equally, in the case of integrated pest management, the sensibility of potential biocontrol strains to chemicals and their adaptability to nursery or field conditions, as well as the optimization of product formulation, are relevant points to be studied.

## Figures and Tables

**Figure 1 jof-09-00638-f001:**
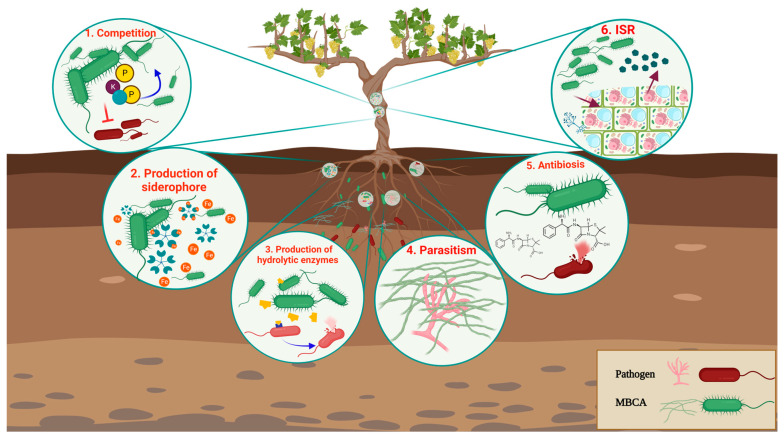
The key mechanisms of action of MBCAs assessed toward GTD-associated fungi. (1) Competition for space and nutrients between the MBCAs and the pathogen(s). In terms of nutrients, they compete for micronutrients such as manganese, specific growth substances (i.e., amino acids), or stimulants for germination (i.e., fatty acids). (2) Production of siderophores that mediate iron competition and lead to reduced pathogen populations. (3) Production of hydrolytic enzymes that permeabilize and degrade the pathogen cell wall (e.g., chitinase, glucanase, protease, and cellulase…), causing cell death. (4) Parasitism, consisting of a direct attack of the pathogen by the MBCA, which leads to the invasion and destruction of the pathogen. (5) Antibiosis, whereby the MBCAs produce inhibitory metabolites or antibiotics that affect the growth or the metabolic activity of the plant pathogen. (6) Induced systematic resistance (ISR), whereby the MBCAs induce a plant defense response similar to that induced after pathogen infection. Created with BioRender.com. Accessed on 20 February 2023.

**Table 1 jof-09-00638-t001:** MBCAs officially registered for the management of GTDs in different countries throughout the world. BPDB: Bio-Pesticides Database, accessible at http://sitem.herts.ac.uk/aeru/bpdb/index.htm (accessed on 24 May 2023); AU: Australia; BE: Belgium; CA: Canada; CY: Cyprus; CZ: Czech Republic; DE: Germany; EL: Greece; ES: Spain; EU: European Union; FR: France; HR: Croatia; HU: Hungary; IT; Italy; KE; Kenya; LU: Luxembourg; MA: Morocco; NL: The Netherlands; NZ: New Zealand; PL: Poland; PT: Portugal; RO: Romania; SA: South Africa; SI: Slovenia; TR: Turkey; UK; United Kingdom; USA: United States of America; VT: Vietnam; ZM: Zambia.

	Trade Name	MBCAs	Mode of Action	Target Pathogen(s)/Disease	Country	References
Fungus	Vintec^®^/Treadani1	*T. atroviride* SC1	Antibiosis; nutrient and space competition; stimulation of plant defenses	*P. chlamydospora*, *P. minimum*, *D. seriata*, *E. lata*, *E. armenicae*, *B. ribis*, and grey mold	BE, CY, CZ, DE, EL, ES, FR, HR, HU, IT, LU, NL, PL, PT, RO, SI, UK, NZ, USA	BPDB [113,114]
	Esquive^®^/Tri-Wall	*T. atroviride I-1237*	Competition for space and nutriments; mycoparasitism	Esca (*Phaeomoniella, Phaeoacremonium*) and botryosphaeria dieback, *E. lata* and also used for the control of root diseases and damping-off	CY, ES, FR, IT, PT, AU, NZ, SA, VT	BPDB [115]
	Eco 77	*T. harzianum* strain B77	Competition for space and nutrients	Eutypa and botrytis	SA; KE, ZM	BPDB [115]
Mix	Blindar/Cassat WP/Remedier^®^/Escalator Bioten WP	*T. asperellum ICC012* & *T. gamsii ICC080*	Antibiosis; mycoparasitism; colonization of pruning wounds; nutrient and space competition	Fungus involved in Esca, Botryosphaeria, and Eutypa dieback Grapevine trunk disease; soil-borne pathogens	USA, CA, EU Members ES, FR, IT, SI), MA, SI, TR	BPDB [115]
	Vinevax Bio-dowel	5 strains of *T. atroviride*	Stimulation of the systemic protective response	Eutypa dieback (*E. lata*) and botryosphaeria dieback (*Botryosphaeria stevensii*)	NZ, AU	[115]
	Vinevax™	5 strains of *T. atroviride*	Competition for space and nutrients	Eutypa dieback (*E. lata*) black dead arm (*Botryosphaeria* spp.) and Petri disease (*P. chlamydospora*).	NZ, AU	[115]

**Table 2 jof-09-00638-t002:** The mechanism of action of the MBCAs against GTD-associated fungi.

MBCAs	Strains	Mechanisms of Action	Targeted Pathogens	References
Fungi				
*Trichoderma*	*T. afroharzianum*	Mycoparasitism	*N. parvum*, *D. seriata*, and *E. lata*	[31]
	*T. asperelloides*	Competition for space	*L. theobromae*, *N. parvum*, *and D. seriata*,	[37,40,62]
	*T. asperellum*	Competition for nutrients and/or space	*P. chlamydospora*, *P. minimum*, *L. theobromae*, and *E. lata*	BPDB, [42,62,72]
	*T. atroviride*	Competition for space and nutrients, production of lytic enzymes, antibiosis, mycoparasitism, and stimulation of plant defense mechanisms	*P. chlamydospora*, *P. minimum*, *D. seriata*, *Botryosphaeria ribis E. lata*, *N. parvum*, *N. australe*, *E. armenicae*, *P. viticola*, *and N. mediterraneaum*	BPDB, [1,33,34,35,37,62,108,113,115]
	*T. canadense*	NA	*N. parvum* and *D. seriata*	[37]
	*T. gamsii*	Antibiosis and mycoparasitism	*P. chlamydospora* and *B. stevenssi*	BPDB, [1]
	*T. guizhouense*	Competition for nutrients and space	*N. parvum*, *D. seriata*, and *E. lata*	[33]
	*T. hamatum*	Competition for space and nutrients	*N. parvum*, *P. chlamydospora*, and *E. lata.*	[42]
	*T. harzianum*	Competition for space and nutrients, mycoparasitism antibiosis, and enhancement of the grapevine defense response	*P. chlamydospora*, *N. parvum*, *D. seriata*, *E. lata*, *P. viticola*, *P. minimum*, *N. australe*, and *L. theobromae*	[1,33,34,35,37,56,62,73]
	*T. koningii*	NA	*P. chlamydospora*, *P. mínimum*, *N. parvum*, and *D. seriata*	[37,81]
	*T. koningiopsis*	Competition for nutrients and space	*N. parvum*, *D. seriata*, *E. lata*, and *L. theobromae*	[33,62]
	*T. longibrachiatum*	Competition for nutrients and space, and enhancement of grapevine defense response	*D. seriata*, *N. parvum*, *E. lata*, and *P. chlamydospora*	[33,56,73]
	*T. paratroviride*	Competition for nutrients and space	*N. parvum*, *D. seriata*, and *E. lata*	[33]
	*T. paraviridescens*	Competition for nutrients and space	*N. parvum*, *D. seriata*, and *E. lata*	[33]
	*T. simmonsii*	Mycoparasitism	*N. parvum*, *D. seriata*, and *E. lata*	[31]
	*T. spirale*	Competition for nutrients and space	*N. parvum*, *D. seriata*, and *E. lata*	[33]
	*T. tomentosum*	NA	*N. parvum* and *D. seriata*	[37]
	*T. viticola*	NA	*N. parvum* and *D. seriata*	[37]
	*Trichoderma* sp.	Competition for nutrients and space, and mycoparasitism	*D. seriata*, *P. chlamydospora*, *P. minimum*, and *E. lata*	[30,33,56,86]
*Epicoccum*	*E. layuense*	Production of diffusible metabolites *in vitro* and competition for space and nutrients	*P. chlamydospora*, *P. minimum*, and *F. mediterranea*	[38]
	*E. mezzettii*	Production of diffusible metabolites *in vitro* and competition for space and nutrients	*P. chlamydospora*, *P. minimum*, and *F. mediterranea*	[38]
	*E. purpurascens*	NA	*L. theobromae*	[63]
*Fusarium*	*F. lateritium*	Antibiosis	*E. lata*	[100,102]
	*F. oxysporum*	Colonization of xylem tissue (competition)	*P. chlamydospora*	[78]
	*F. proliferatum*	Direct antagonism (antibiosis) and priming plant defense response	*N. parvum* and *D. seriata*	[32]
*Cladosporium*	*C. herbarum*	The colonization of pruning wounds by its hydrophobic conidia (completion)	*E. lata*,	[102]
	*Cladosporium* sp.	Antibiosis and high rate of sporulation (competition)	*N. parvum*, *D. seriata*, and *P. chlamydospora*	[30]
*Aureobasidium*	*A. pullulans*	Direct antagonism (stopped growth)	*N. parvum*, *D. seriata*, and *E. lata*	[42,102]
*Candida*	*C. famata*	NA	*E. lata*	[102]
*Chaetomium*	*Chaetomium* sp.	Mycoparasitism	*N. parvum*, *D. seriata*, and *P. chlamydospora*	[30]
*Clonostachys*	*C. rosea*	Antibiosis and mycoparasitism	*D. seriata*, *N. parvum*, *P. chlamydospora*, *P. mínimum*, and *E. lata*	[30,40]
*Lecanicillium*	*L. lecanii*	Competition for space and nutrients	*N. parvum, D. seriata*, *P. chlamydospora*, *P. minimum*, and *E. lata*	[35]
*Penicillium*	*Penicillium* sp.	NA	*E. lata*	[102]
*Purpureocillium*	*P. lilacinum*	Direct antagonism (secreted secondary metabolites)	*N. parvum*, *D. seriata*, and *P. chlamydospora*	[30]
*Rhodotorula*	*R. rubra*	NA	*E. lata*	[102]
Bacteria				
*Achromobacter*	*Achromobacter* sp.	NA	*F. mediterranea*	[71]
*Bacillus*	*B. amyloliquefaciens*	Antibiosis	*L. theobromae*, *P. chlamydospora*, and *P. minimum*	[65]
	*B. cereus*	Direct antagonism	*E. lata*	[103]
	*B. firmus*	NA	*N. parvum* and *P. chlamydospora*	[81]
	*B. licheniformis*	Direct antagonism	*P. minimum*	[89]
	*B. methylotrophicus*	Direct antagonism	*N. parvum*, *P. chlamydospora*, and *P. minimum*	[49]
	*B. pumilus*	Induction of the expression of defense-related genes	*N. parvum* and *P. chlamydospora*	[52,53,82]
	*B. subtilis*	Antibiosis and induction of the expression of defense-related genes	*N. parvum*, *D. seriata*, *L. theobromae*, *N. australe*, *P. chlamydospora*, *P. minimum*, and *E. lata*	[28,52,82,101,103,104]
	*B. thuringiensis*	Antibiosis and competition for nutrient	*E. lata*	[103]
	*B. velezensis*	NA	*N. parvum*, *D. seriata*, *L. theobromae*, *P. minimum*, and *E. lata*	[42,51]
	*Bacillus* sp.	NA	*P. chlamydospora* and *F. mediterranea*	[52,71]
*Brevibacillus*	*B. reuszeri*	NA	*P. chlamydospora*	[82]
	*Brevibacillus* sp.	NA	*N. parvum*	[49]
*Brevundimonas* sp.	*Brevundimonas* sp.	NA	*F. mediterranea*	[71]
*Burkholderia*	*Burkholderia* sp.	NA	*F. mediterranea*	[71]
*Cedecea* sp.	*Cedecea* sp.	NA	*F. mediterranea*	[71]
*Chryseobacterium*	*Chryseobacterium* sp.	NA	*F. mediterranea*	[71]
*Curtobacterium*	*Curtobacterium* sp.	NA	*F. mediterranea*	[71]
*Enterobacter*	*Enterobacter* sp.	NA	*N. parvum*, *P. chlamydospora*, and *F. mediterranea*	[71,82]
*Frigoribacterium*	*Frigoribacterium* sp.	NA	*F. mediterranea*	[71]
*Erwinia*	*Erwinia* sp.	NA	*F. mediterranea*	[71]
*Herbiconiux*	*Herbiconiux* sp.	NA	*F. mediterranea*	[71]
*Kocuria*	*Kocuria* sp.	NA	*F. mediterranea*	[71]
*Luteimonas*	*Luteimonas* sp.	NA	*F. mediterranea*	[71]
*Lysinibacillus*	*Lysinibacillus* sp.	NA	*F. mediterranea*	[71]
*Microbacterium*	*Microbacterium* sp.	NA	*F. mediterranea*	[71]
*Novosphingobium*	*Novosphingobium* sp.	NA	*F. mediterranea*	[71]
*Olivibacter*	*Olivibacter* sp.	NA	*F. mediterranea*	[71]
*Paenibacillus*	*P. alvei*	NA	*P. chlamydospora*	[78]
	*P. illinoisensis*	NA	*P. chlamydospora*	[82]
	*Paenibacillus* sp.	Induction of the expression of defense-related genes and antibiosis	*N. parvum* and *P. chlamydospora*	[22,52,82]
*Pseudomonas*	*P. protegens*	NA	*N. parvum*, *P. minimum*, and *P. chlamydospora*	[50]
	*P. fluorescens*	NA	*E. lata*	[102,103]
	*P. chlororaphis*	NA	*N. parvum*, *L. theobromae*, *P. minimum*, and *E. lata*	[51]
	*P. aeruginosa*	NA	*E. lata*	[103]
	*Pseudomonas* sp.	NA	*F. mediterranea*, *N. parvum*, *and D. seriata*	[51,71]
*Pantoea*	*P. agglomerans*	Induction of the expression of defense-related genes, antibiosis, and production of siderophores	*N. parvum*, *P. chlamydospora*, *E. lata*, and *F. mediterranea*	[52,53,71,82,103]
*Pedobacter*	*Pedobacter* sp.	NA	*F. mediterranea*	[71]
*Pigmentifaga*	*Pigmentifaga* sp.	NA	*F. mediterranea*	[71]
*Pseudoxanthomonas*	*Pseudoxanthomonas* sp.	NA	*F. mediterranea*	[71]
*Rahnella*	*Rahnella* sp.	NA	*F. mediterranea*	[71]
*Rhizobiaceae*	*/*	NA	*F. mediterranea*	[71]
*Serratia*	*S. plymuthica*	NA	*E. lata*, *N. parvum*, *D. seriata*, *L. theobromae*, and *P. mínimum*	[51,103]
*Sphingomonas*	*Sphingomonas* sp.	NA	*F. mediterranea*	[71]
*Stenotrophomonas*	*S. maltophilia*	NA	*E. lata*	[103]
	*Stenotrophomonas* sp.	NA	*F. mediterranea*	[71]
*Variovorax* sp.	*Variovorax* sp.	NA	*F. mediterranea*	[71]
*Xanthomonaceae*	*/*	NA	*F. mediterranea*	[71]
Actinobacteria				
*Streptomyces*	*Streptomyces* sp.	NA	*P. chlamydospora*, *P. minimum*, *N. parvum*, and *E. lata*	[54,81,103]
Oomycete				
*Pythium*	*P. oligandrum*	Induction of systemic resistance	*N. parvum* and *P. chlamydospora*,	[43,53,55,79,80]

NA: not available.

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
