# Peer review of "Microbial Biological Control of Fungi Associated with Grapevine Trunk Diseases: A Review of Strain Diversity, Modes of Action, and Advantages and Limits of Current Strategies"

_jof, 2023, doi:10.3390/jof9060638_

Round 1
Reviewer 1 Report (Previous Reviewer 1)
The manuscript has been greatly improved.
Lowercase letters of “Grapevine Trunk Diseases” should be used in title and text.
Author Response
Response to Reviewer 1
Thank you for revising our review.
“Lowercase letters of “Grapevine Trunk Diseases” should be used in title and text.”
This comment has been taken into account and we have changed it in the text.

Reviewer 2 Report (New Reviewer)
The work fills all the requirements of the review work.
The authors use relatively new literature and try to familiarize the reader with the discussed issues in a fairly orderly way.
However:
Please standardize the writing scheme. The work gives the impression of being written by two separeted people without taking into account the corrections between them.
In addition, the work is clearly unworked in terms of final perfection.
More specific comments:
The authors try to demonstrate the legitimacy of the use of specific type of plant biological protection techniques
The publication systematizes knowledge in this field
It provides a comprehensive overview of the available literature
Methodology does not need improvement
The conclusions and summary are clear and correspond to the issues raised
Literature selection correct
Tables and drawings do not need to be improved.
Author Response
Response to Reviewer 2
Thank you for revising our review.
“Please standardize the writing scheme. The work gives the impression of being written by two separated people without taking into account the corrections between them.”
The review was written and revised by the authors and we have tried to harmonize the writing style as you can see in the new version.
“In addition, the work is clearly unworked in terms of final perfection.”
This comment has been taken into account, we have corrected all the typing errors, and changed the text formatting when it was required.

This manuscript is a resubmission of an earlier submission. The following is a list of the peer review reports and author responses from that submission.
Round 1
Reviewer 1 Report
This review introduced the category, modes of action, efficacy, and limits of biocontrol microorganisms against grapevine trunk diseases caused by fungi. It is quite informative; however, the current version is very coarse with the stack of original references, it needs to be well organized and refined.
1. “Grapevine Trunk Diseases-fungi” in the title is inaccurate and not clear.
2. This review focused on microbial biocontrol of fungal GTDs, so I would suggest in the first part of Instruction, just introduce distribution and damage of the diseases, causing pathogens, as well as control practices currently used and their limits. The defense mechanisms of grapevine are far from the topic and they are universal in plants, so these contents including Figure 1 should be cancelled.
3. The arrangement of the text is a bit of a mess. It’s not necessary to use so many levels and emphasized phrases and sentences; the use of explicit subtitles with a main thread is fine. To describe the biocontrol of different pathogenic fungi, it’s clear to make outline with biocontrol species instead of assay methods or both, for many of the isolates were tested both in vitro and in planta, with consistent efficacy. For oomycete, it is more conforming to put it ahead of bacteria and antinomyces.
4. I would suggest to combine the mechanisms of action in Introduction with the corresponding part in Section 4.
5. Separate Section 4 “Mechanisms of action, Efficiency, durability, and resistance to MBCAs” into two parts: “Mechanisms of action of MBCAs against GTDs” and “Influence factors on control efficiency of the MBCAs”.
6. For a high level review, neat subtitles are required. I would suggest to improve the Heading level 2 ( 2.1. Biocontrol of Botryosphaeria dieback, 2.2. Esca: the main worldwide developed grapevine trunk disease, 2.3. Eutypa dieback).
7. Trichoderma spp. are the most commonly used microbial fungicides against various plant fungal diseases in all over the world; the functions of Trichoderma commercial products to control other soilborne and airborne diseases should not be included in this manuscript. Moreover, Table 1 seemed very crowd, some information can be shown in the text, and the explanations should be put under the table as a note.
8. The description of resistance of field pests to bioinsecticide should be cancelled, for it has nothing to do with pathogen’s influence (Line 1295-1298).
9. Change “GTDs associated” to “GTD-associated”.
10. Latin names of the pathogens and biocontrol microorganisms, and specific terms such as in vitro, should be written in italic in subtitles and whole text, while sp. and spp. regularly scripted all the time.
Minor mistakes:
There is only 2.3.1.
Line 105-106, Change to “These MBCAs play important roles in”.
Line 107-112, I would suggest to rearrange the sentences, i.e., introduce direct interaction first, then indirect interaction.
Line 142, Change “induce” to “cause” or “lead to”.
Formats and spellings should be carefully checked, e.g. Caption of Figure 2, Line 2: strains diversity, Line 144: as?, Line 797: funganl strains……
Author Response
Response to Reviewer 1:
- “Grapevine Trunk Diseases-fungi” in the title is inaccurate and not clear.
We have considered your comment and suggest this new title: “Microbial biocontrol of fungi associated with Grapevine Trunk Diseases: strains diversity, mode of actions, advantages and limits of current strategies. A review”
- “This review focused on microbial biocontrol of fungal GTDs, so I would suggest in the first part of Instruction, just introduce distribution and damage of the diseases, causing pathogens, as well as control practices currently used and their limits”.
- Regarding “The distribution and damage of the diseases and causing pathogens”: in the section "lines 37 to 54", we described GTDs in general and cited pathogens associated with these diseases. Besides, in each section for the three major diseases (i.e. Esca, Botryosphaeria dieback and Eutypa dieback), we gave details for each disease and the associated pathogens.
- Regarding the “control practices currently used and their limits”: we have taken into account this comment and add a new paragraph on this topic (lines 98 to 141).
- “The defense mechanisms of grapevine are far from the topic and they are universal in plants, so these contents including Figure 1 should be cancelled.”The sub-section and the corresponding figure were removed.
- The arrangement of the text is a bit of a mess. It’s not necessary to use so many levels and emphasized phrases and sentences; the use of explicit subtitles with a main thread is fine. We have taken into account this comment and we re-organized the text and reviewed the subtitles. To give more understandability and clarity, at the end of the sections dedicated to the control of the major GTD-associated pathogens, we added some comments comparing the studies obtained in vitro and, on the plants, (e.g. in planta, in the greenhouses, in the vineyards). (Line 434 to 445; line 557 to 564; line 608 to 613; line 870 to 876; line 959 to 965; line 996 to 1000; line 1208 to 1216).
- For oomycete, it is more conforming to put it ahead of bacteria and For each GTD disease, we moved the oomycete section in the text (line 753 to 772 and line 923 to 925) before the section about bacteria and actinomyces.
- I would suggest to combine the mechanisms of action in Introduction with the corresponding part in Section 4. We have separated this section from section 4 in order to introduce the different MBCAs mechanisms of action that inhibit the GTD-associated pathogens (lines 150 to 155). The figure was used to describe each mechanism, our purpose being to define each mechanism before mentioning them later in the different sections.
- Separate Section 4 “Mechanisms of action, Efficiency, durability, and resistance to MBCAs” into two parts: “Mechanisms of action of MBCAs against GTDs” and “Influence factors on control efficiency of the MBCAs”. We have separated the section 4 into two sections.
- For a high-level review, neat subtitles are required. I would suggest to improve the Heading level 2 ( 2.1. Biocontrol of Botryosphaeria dieback, 2.2. Esca: the main worldwide developed grapevine trunk disease, 2.3. Eutypa dieback). We have reviewed paragraph titles and we propose: 2.1. Biocontrol of fungi associated with Botryosphaeria dieback; 2.2. Biocontrol of fungi associated with Esca; 2.3. Biocontrol of fungi associated with Eutypa dieback.
- Trichoderma are the most commonly used microbial fungicides against various plant fungal diseases in all over the world; the functions of Trichoderma commercial products to control other soilborne and airborne diseases should not be included in this manuscript. These products are used to treat several fungal diseases but here, we have mentioned them because they are used in several countries to control GTDs.
- Table 1 seemed very crowd, some information can be shown in the text, and the explanations should be put under the table as a note. We have removed the formulation and their use from the table 1 which are now mentioned in the text.
- The description of resistance of field pests to bioinsecticide should be cancelled, for it has nothing to do with pathogen’s influence (Line 1295-1298).
It was mentioned as an example to indicate that the development of resistance to biological control agents has been studied for some MBCAs. That’s why we decided to keep this mention to Bacillus thuringiensis even if it was not in the context of the GTDs because so far, there is no study that focused on pathogen fungal resistance to MBCAs.
- Change “GTDs associated” to “GTD-associated”. We changed “GTDs associated” to “GTD-associated.
- Latin names of the pathogens and biocontrol microorganisms, and specific terms such as in vitro, should be written in italic in subtitles and whole text, while sp. and spp. regularly scripted all the time. The whole text has been revised and we wrote the Latin names in italic.
Minor mistakes:
- There is only 2.3.1. we checked the subtitles.
- Line 105-106, Change to “These MBCAs play important roles in”. we have changed this in the text.
- Line 107-112, I would suggest to rearrange the sentences, i.e., introduce direct interaction first, then indirect interaction. It is in this order in the text.
- Line 142, Change “induce” to “cause” or “lead to”. We changed the sentence in the text.
- Formats and spellings should be carefully checked, e.g. Caption of Figure 2, Line 2: strains diversity, Line 144: as? Line 797: fungal strains…… These points were checked
Reviewer 2 Report
The text is extremely long and dispersive. Several paragraphs are repetitive and the content could be summarized in a table. Many references are incorrect, for example line 142 (refence 25) the article by Bahmani et al., concerns Biscogniauxia rosacearum and not species of Botryosphaeriaceae. At the same time several references are incomplete (eg, 37, 59, 65……..). All scientific names should be in italics, please check the manuscript carefully.
Below some edits aimed to improve the manuscript:
Line 138: the number of Botryosphaeriaceae species associated with grapevine diseases is greater than 26, please check the literature.
Line 496: many species identified as Lasiodiplodia theobromae actually belong to other species such as L. pseudotheobromae and L. mediterranea, this paragraph should be rewritten.
Line 554: please use the current name for the species: Botryosphaeria stevensii should be replaced with Diplodia mutila.
The conclusion should be improved, after 35 pages the content of the conclusion is really poor.
Author Response
Response to Reviewer 2:
“Many references are incorrect, for example line 142 (refence 25) the article by Bahmani et al., concerns Biscogniauxia rosacearum and not species of Botryosphaeriaceae.” We have double-checked this point and we confirmed that Bahmani et al. isolated strains of P. minimum associated with Botryosphaeria dieback.
“At the same time several references are incomplete (eg, 37, 59, 65……...).” We checked and completed all the references.
“All scientific names should be in italics, please check the manuscript carefully.” We have revised the whole text and we wrote the scientific names in italic.
Below some edits aimed to improve the manuscript:
- Line 138: the number of Botryosphaeriaceae species associated with grapevine diseases is greater than 26, please check the literature. We checked the literature and we changed the word “species” by “taxa”. Indeed, in the article from Bettenfeld et al [1], they mentioned around “30” taxa are associated to Botryosphaeria dieback, 21 taxa in the publications of Bertsch et al. and Úrbez-Torres [2,3] and 26 taxa in the article of Gramaje et al. [4].
- Line 496: many species identified as Lasiodiplodia theobromae actually belong to other species such as L. pseudotheobromae and mediterranea, this paragraph should be rewritten. We noted in the text that the taxonomy for L. theobromae has recently been revised [5]. However, we can’t change the taxonomy in our text because original publications mentioned only the former identification.
- Line 554: please use the current name for the species: Botryosphaeria stevensii should be replaced with Diplodia mutila. We have taken in account this comment and we changed the names of the species.
- The conclusion should be improved, after 35 pages the content of the conclusion is really poor. We have combined the sections "outlooks" and "Conclusion”, because they actually have the same purpose. This new section is reorganized and enriched with new ideas.
- Bettenfeld, P.; Cadena i Canals, J.; Jacquens, L.; Fernandez, O.; Fontaine, F.; van Schaik, E.; Courty, P.-E.; Trouvelot, S. The Microbiota of the Grapevine Holobiont: A Key Component of Plant Health. Journal of Advanced Research 2022, 40, 1–15, doi: 10.1016/j.jare.2021.12.008.
- Bertsch, C.; Ramírez-Suero, M.; Magnin-Robert, M.; Larignon, P.; Chong, J.; Abou-Mansour, E.; Spagnolo, A.; Clément, C.; Fontaine, F. Grapevine Trunk Diseases: Complex and Still Poorly Understood: Grapevine Trunk Diseases. Plant Pathology 2013, 62, 243–265, doi:10.1111/j.1365-3059.2012.02674.x.
- 5438-Article Text-5366-1-1-20190717.Pdf.
- Gramaje, D.; Úrbez-Torres, J.R.; Sosnowski, M.R. Managing Grapevine Trunk Diseases with Respect to Etiology and Epidemiology: Current Strategies and Future Prospects. Plant Disease 2018, 102, 12–39, doi:10.1094/PDIS-04-17-0512-FE.
- Du, Y.; Wang, X.; Guo, Y.; Xiao, F.; Peng, Y.; Hong, N.; Wang, G. Biological and Molecular Characterization of Seven Diaporthe Species Associated with Kiwifruit Shoot Blight and Leaf Spot in China. Phytopathol. Mediterr. 2021, 60, 177–198, doi:10.36253/phyto-12013.
Round 2
Reviewer 1 Report
1. The authors added several paragraphs to explain cultural practices controlling GTDs. In the review, a brief introduction of the current situation should be included; however, much information of specific technique and detailed operation methods may interfere the main purpose of the article.
2. The description of resistance of field pests to bioinsecticide should be cancelled, for this review dose not aim to display all mechanisms underlying MBCAs, so just indicate those related to the biocontrol of GTDs.
3. Arrangement of the biocontrol of 3 GTDs is still some mess; the highest of fifth-grade subtitles are used, which is much complex. I would suggest to cancel heading level 2 “GTD-associated pathogens and biocontrol”, instead, using “Biocontrol of Botryosphaeria dieback”, “Biocontrol of Esca” and “Biocontrol of Eutypa lata” as Heading level 1. Moreover, biocontrol microbe species should be main clue then the next level of in vitro and in planta assays, but not mixed together. For example, in sections “Bacterial biocontrol of Botryosphaeria dieback pathogens has been---”, “Other fungal genera evaluated to control N. parvum”, etc., experiments in vitro and in planta should follow the given species.
4. In general, the term “biocontrol” is more linked with a disease, but not a pathogen.
5. In the last part of the review, more challenges and prospects were discussed instead of “conclusion”.
Author Response
Response to Reviewer 1
- The authors added several paragraphs to explain cultural practices controlling GTDs. In the review, a brief introduction of the current situation should be included; however, much information of specific technique and detailed operation methods may interfere the main purpose of the article.
We summarized the section which describes the current strategies used to control GTDs by cultural practices.
In the lines 90 to 96 we describe the current situation regarding the control of these diseases
- The description of resistance of field pests to bioinsecticide should be cancelled, for this review dose not aim to display all mechanisms underlying MBCAs, so just indicate those related to the biocontrol of GTDs. We deleted this section in the manuscript.
- Arrangement of the biocontrol of 3 GTDs is still some mess; the highest of fifth-grade subtitles are used, which is much complex. I would suggest to cancel heading level 2 “GTD-associated pathogens and biocontrol”, instead, using “Biocontrol of Botryosphaeria dieback”, “Biocontrol of Esca” and “Biocontrol of Eutypa lata” as Heading level 1. Moreover, biocontrol microbe species should be main clue then the next level of in vitro and in planta assays, but not mixed together. For example, in sections “Bacterial biocontrol of Botryosphaeria dieback pathogens has been---”, “Other fungal genera evaluated to control N. parvum”, etc., experiments in vitro and in planta should follow the given species.
We have changed the arrangement of the paragraphs and the level of titles and the subtitles according to the guidelines you suggested.
Moreover, biocontrol microbe species should be main clue then the next level of in vitro and in planta assays, but not mixed together. For example, in sections “Bacterial biocontrol of Botryosphaeria…
We have removed mention to in planta/in vitro in the titles, we have homogenized all the titles for all the diseases. In this version we reach, at maximum 3 sub-levels. For instance, for the first section after introduction we have now this structure:
2.Biocontrol of Botryosphaeria dieback
2.1. Biological control of Neofusicoccum parvum
2.1.1 Biocontrol using Trichoderma
2.1.2 Biocontrol using other fungal genera
2.1.3 Biocontrol using bacteria
2.1.4. Biocontrol using actinobacteria
- In general, the term “biocontrol” is more linked with a disease, but not a pathogen. We have taken this comment in account and we changed the “Biocontrol of fungi associated with…” to “Biocontrol of …” In the titles of the different sections. And we changed “biocontrol of “pathogen” by biological control of “pathogen” in the text.
- In the last part of the review, more challenges and prospects were discussed instead of “conclusion”.
We divided this section into two parts: with the following titles:
- General discussion: challenges and prospects: line 1449 to 1496.
- Conclusion de depart: line 1497 to 1527.
Our main issue here is that ‘Reviewer 2’ wanted us to write a longer and dense conclusion. So, we merged the section ‘general discussion’ that include challenges and prospects and the ‘conclusion’. However, based on your remark, we separated the 2 sections. In our opinion, the conclusion cannot be too long and perspectives has to be separated from the true conclusion.
Reviewer 2 Report
The manuscript has not been improved, several issues including bibliographic references have not been corrected adequately and some of the new sentences inserted refer to incorrect references e.g. line 569 the new sentence: “The taxonomy of Lasiodiplodia was recently revised. As a consequence, fungal isolates previously reported as L. theobromae were reclassified as new species. A number of species were then reduced to synonyms [62].” However, the reference 62 refers to a manuscript about Diaporthe species:
Du, Y.; Wang, X.; Guo, Y.; Xiao, F.; Peng, Y.; Hong, N.; Wang, G. Biological and molecular characterization of seven Diaporthe species associated with kiwifruit shoot blight and leaf spot in China. Phytopathol. Mediterr. 2021, 170360, 177–198, doi:10.36253/phyto-12013
Author Response
Response to Reviewer 2
The manuscript has not been improved, several issues including bibliographic references have not been corrected adequately and some of the new sentences inserted refer to incorrect references e.g. line 569 the new sentence: “The taxonomy of Lasiodiplodia was recently revised. As a consequence, fungal isolates previously reported as L. theobromae were reclassified as new species. A number of species were then reduced to synonyms [62].” However, the reference 62 refers to a manuscript about Diaporthe species:
We did our best to improve the manuscript combining both reviewers recommendations. We apologize for the mistake remaining in one reference. This is now modified. We have double checked all the references and we are confident now that everything is fine. Please give use more specific comments if you have identified something else.